



# Investigation of the wet removal rate of black carbon in East Asia: validation of a below- and in-cloud wet removal scheme in FLEXPART v10.4

Yongjoo Choi[1], Yugo Kanaya[1], Masayuki Takigawa[1], Chunmao Zhu[1], Seung-Myung Park[2], Atsushi Matsuki[3], Yasuhiro Sadanaga[4], Sang-Woo Kim[5], Xiaole Pan[6], Ignacio Pisso[7]

[1] Research Institute for Global Change, Japan Agency for Marine-Earth Science and Technology (JAMSTEC), Yokohama, 2360001, Japan

[2] Division of Climate & Air Quality Research, National Institute of Environmental Research, Kyungseo-dong, Seo-Gu, Incheon 404170, Korea

[3] Institute of Nature and Environmental Technology, Kanazawa University, Kanazawa 9201192, Japan

[4] Department of Applied Chemistry, Graduate School of Engineering, Osaka Prefecture University, 1-1 Gakuen-cho, Naka-ku, Sakai, Osaka 5998531, Japan

[5] School of Earth and Environmental Sciences, Seoul National University, Seoul 08826, Korea

[6] Institute of Atmospheric Physics, Chinese Academy of Sciences, Beijing, China

[7] NILU – Norwegian Institute for Air Research, Kjeller 2027, Norway

*Correspondence to:* Yongjoo Choi (choingjoo@jamstec.go.jp)

**Abstract**

Understanding the global distribution of atmospheric black carbon (BC) is essential to unveil its climatic effect. However, there are still large uncertainties regarding the simulation of BC transport due to inadequate information about the removal process. We accessed the wet removal rate of BC in East Asia based on long-term measurements over the 2010–2016 period at three representative background sites (Baengnyeong and Gosan in South Korea and Noto in Japan). The average wet removal rate, represented by transport efficiency (TE), i.e. the fraction of undeposited BC particles during transport, was estimated as 0.73 in East Asia from 2010 to 2016. According to accumulated precipitation along trajectory, TE was lower in East and North China, where the industrial sector (thin-coated) is dominant; in contrast, that in South Korea and Japan showed higher values due to the transport sector (thick-coated), with emissions mainly from diesel vehicles. By the same token, TE in winter and summer showed the highest and lowest values, respectively, depending on the dominant emission sectors, such as house heating (thick-coated) and industry. The average half-life and *e*-folding lifetime of BC were 2.8 and 7.1 days, respectively, similar to previous studies, but those values differed according to the geographical location and meteorological conditions of each site. Next, by comparing TE from the FLEXible PARTicle (FLEXPART) Lagrangian transport model (version 10.4), we diagnosed the scavenging coefficients ($s^{-1}$) of the below- and in-cloud scavenging scheme implemented in FLEXPART. The overall median TE from FLEXPART (0.91) was overestimated compared to the measured value, implying underestimation of wet scavenging coefficients in the model simulation. The median of the below-cloud scavenging coefficient showed a lower value than that calculated from FLEXPART, by a factor of 1.7. On the other hand, the overall median of the FLEXPART in-cloud scavenging coefficients was highly underestimated by 1 order of magnitude compared to the measured value. From the analysis of artificial neural networks, the convective available potential energy, which is well known as an indicator of vertical instability, should be considered in the in-cloud scavenging process to improve the representative regional difference in BC wet scavenging over East Asia. For the first time, this study suggested an effective and straightforward evaluation method for wet scavenging schemes (both below- and in-cloud) by introducing TE along with excluding effects from the inaccurate emission inventories.



## 1. Introduction

Black carbon (BC) is the most significant light-absorbing aerosol that can cause positive radiative forcing on climate change (Winiger et al., 2016; Myhre et al., 2013; Bond et al., 2013; Emerson et al., 2018). However, state-of-the-art models still have a limitation in evaluating the direct radiative forcing of BC because of the large model uncertainties in simulating BC concentrations (Xu et al., 2019; Bond et al., 2013; Samset et al., 2014; Wang et al., 2014a). This can partly be attributed to the following three reasons: inaccurate bottom-up emission inventory, the complexity of BC hygroscopicity, and an imprecise dry/wet deposition scheme. First, because BC is mainly contributed by scattered emission sources, the uncertainty of BC emission rates is large compared to other species whose emissions are dominated by large sources (Zheng et al., 2018). Second, BC itself is hydrophobic immediately after emission, is subsequently converted to possessing hydrophilic properties through the aging process and transportation (Moteki et al., 2007; Matsui et al., 2018), and finally acts as cloud condensation nuclei (Kuwata et al., 2007; Bond et al., 2013). Third, while BC particles are transported in the atmosphere, they can be removed by dry and/or wet deposition, including below-cloud (i.e., washout) and in-cloud (i.e., rainout) processes. Wet deposition is still challenging to predict BC concentration in the atmosphere due to the difficulties of accurate evaluation of wet removal (Emerson et al., 2018; Bond et al., 2013; Lee et al., 2013). Specifically, the in-cloud process is more efficient and complicated than the below-cloud process because the nucleation removal of aerosol particles within clouds is thought to account for more than 50% of the aerosol particle mass removal from the atmosphere globally (Grythe et al., 2017; Textor et al., 2006).

Accompanied with the refinement of BC emission inventories over East Asia (Choi et al., 2020; Kanaya et al., 2016), wet removal rates have been one of the main topics to better predict BC behavior by using the term transport efficiency (TE), which is the fraction of undeposited BC particles during transport, because TE has been proven to be a good proxy for wet scavenging. Moteki et al. (2012), which was further elaborated from Oshima et al. (2012), reported the first observational evidence of the size-dependent activation of BC removal over the Yellow Sea during the Aerosol Radiative Forcing in East Asia (A-FORCE) airborne measurement campaign in the spring of 2009. Kondo et al. (2016) demonstrated an altitude dependence, with typical decreasing size distributions at higher altitudes associated with wet removal from A-FORCE in winter 2013. Kanaya et al. (2016) elucidated the relationship between the wet removal rate of BC and accumulated precipitation along trajectory (APT) from long-term measurements (2009–2015) at Fukue, Japan. Miyakawa et al. (2017) reported the effects of BC aging related to in-cloud scavenging during transport on the alteration of the BC size distribution and mixing stats during the spring of 2015 at the same location. Matsui et al. (2013) demonstrated that the difference in the coating thickness of BC particles depended on the growing process (condensation and coagulation), indicating that the coagulation process is necessary to produce thickly coated BC particles that are preferentially removed via the wet scavenging process. Recently, numerous fine mode particles, including BC from polluted areas scavenging in clouds were more pronounced in East Asia, not only at a local scale but also at a large regional scale (Liu et al., 2018), because high aerosol loading conditions are usually associated with significant cloud cover (Eck et al., 2018).

BC and carbon monoxide (CO) are byproducts of the incomplete combustion of carbon-based fuels, and the ratio between $\Delta BC$ (the difference from the baseline level) and $\Delta CO$ could be a useful parameter for characterizing combustion types. Adopting APT, a useful index for the strength of wet deposition (Kanaya et al., 2016; Kanaya et al., 2019), the magnitude of the BC wet removal rate according to precipitation can be easily characterized by TE. Although some previous studies have investigated wet scavenging schemes in models (Grythe et al., 2017; Croft et al., 2010), those results may include bias due to the effect of inaccurate emission rate because emission rates and deposition terms were not necessarily separated. For the first





time, the emission and deposition terms are distinctly separated in this study by introducing TE; this allows for the wet
scavenging scheme to be evaluated more accurately. By elaborating the regional ΔBC/ΔCO ratio (Choi et al., 2020), this study
investigates the characteristics of the BC wet removal rate over East Asia using long-term measurements (more than 3 years)
with the best effort to acquire reliable BC concentrations with wide spatial coverages over East Asia. The differences in wet
removal rates depending on measurement sites and administrative districts (and season) are discussed in Sections 3.1 and 3.2,
respectively. Afterward, to evaluate the representativeness of the scavenging scheme in the recently updated FLEXible
PARTicle dispersion model (FLEXPART) version 10.4, the wet scavenging coefficients for below- and in-cloud processing
were estimated from the wet removal rate by allocating the air mass location (such as below or within the cloud) and
meteorological variables along the pathway of airmass transport.
**2. Methods**
**2.1 Measurement sites and instruments**
To investigate wet removal rates of the outflow airmass from China and Korea peninsula, BC and CO data from three
measurement sites (Baengnyeong, Gosan in Korea and Noto in Japan; Figure 1a) were carefully selected for this study by
considering major emission sources near the measurement sites and by obtaining reliable BC concentrations from different
instruments. As detailed information on the measurement sites and instruments is described in Choi et al. (2020), we only
address brief information here. Baengnyeong (124.63ºE, 37.97ºN), one of the intensive measurement stations operated by the
Korean Ministry of Environment, is frequently affected by airmasses from China (East, North, and Northeast) and North Korea.
Gosan (126.17ºE, 33.28ºN) is located in the southern part of Korea and is frequently affected by airmasses from East China
and South Korea. BC and CO were also measured at the Noto Ground-based Research Observatory (NOTOGRO, 137.36ºE,
37.45ºN), located on the Noto Peninsula on the western coast of Japan, which is frequently affected by airmasses from
Northeast China and Japan. The measurement periods were mainly in the early 2010s but slightly different depending on the
sites (Figure S1). The longest measurement period was in Noto for approximately 6 years (from 2011 to 2016), followed by
that in Baengnyeong (5 years) and Gosan (3 years).
As the best effort to obtain reliable BC concentrations from different instruments, only well-validated instruments were used
in this study. Hourly $PM_{2.5}$ elemental carbon (EC) was measured by a Sunset EC/OC analyzer with optical correction for
Baengnyeong. Multi-angle absorption photometer (MAAP 5012) was used to measure hourly BC in $PM_{2.5}$ for Noto. At Gosan,
BC in $PM_1$ was monitored by a continuous light absorption photometer (CLAP) with three wavelengths including 467, 528,
and 652 nm and the absorption was corrected following Bond (1999). At Noto, an improved mass absorption efficiency (MAE)
of 10.3 $m^2 g^{-1}$ instead of the default value (6.6 $m^2 g^{-1}$) was applied to estimate the BC mass concentration, as suggested based
on calibrations using the thermal/optical method and the laser-induced incandescence technique (Kanaya et al., 2013; Kanaya
et al., 2016). CLAP also showed a good correlation with co-located $PM_{2.5}$ EC concentration from the Sunset EC/OC analyzer
and the best-fitted line was close to one (1.17), similar or slightly lower than the range of reported uncertainty, ~25% (Ogren
et al., 2017). Hourly CO concentrations were measured by a gas filter correlation CO analyzer (Model 300 EU Teledyne Inc.)
at Baengnyeong and by a nondispersive infrared absorption photometer (48C, Thermo Scientific) at the other two sites. The
overall uncertainty of BC and CO measurements from different instruments was estimated to be less than 15% (except for
Gosan; 20%) and 5%, respectively, which leads to 10% uncertainty of overall regional ΔBC/ΔCO ratio (Choi et al., 2020).





**2.2 Backward trajectory and meteorological data**
To identify the airmass origin region, 72 h backward trajectories were calculated four times a day (00, 06, 12, 18 UTC) using
the Hybrid Single Particle Lagrangian Integrated Trajectory (HYSPLIT) Model version 4 (Draxler et al., 2018). The starting
altitude was 500 m above ground level (AGL). Notably, we used the European Centre for Medium-Range Weather Forecasts
(ECMWF) ERA5, which provides a much finer resolution of 0.25°×0.25°, as inputs for HYSLPIT instead of Global Data
Assimilation System (GDAS) 1°×1° data with 23 pressure levels to improve the accurate assessment of airmass transportation
pathways and to acquire more detailed information on meteorological conditions. According to the pathway of airmass
transportation, the detailed meteorological information for precipitation and clouds was acquired from ERA5 hourly data on
both single and pressure levels (37 levels; 1000 hPa to 1 hPa) to identify the below- and/or in-cloud cases and to calculate the
wet scavenging coefficients.
If precipitation occurred before the airmass arrived at the main BC source region, it is difficult to investigate the wet removal
effect because the effects of precipitation could be underestimated at receptor sites. Therefore, we considered the residence
time (Li et al., 2014; Ashbaugh et al., 1985) of each grid cell (0.25°×0.25°) and the BC emission rates (mass/time) from the
Regional Emission inventory in ASia (REAS; Figure 1a) emission inventory (Kurokawa et al., 2013) version 2.1 to identify
the potential emission region by multiplying residence time and emission rates. First, when the airmass altitude was lower than
2.5 km, the airmass velocities ($V_n$ and $V_{n+1}$) were calculated by distances from the central point in a target grid cell to two-way
endpoints of backward trajectories ($D_n$ and $D_{n+1}$) using $V_n=D_n/\Delta t$ and $V_{n+1}=D_{n+1}/\Delta t$ (Figure 1b). Here, $\Delta t$ and $n$ are the time
interval of meteorological data (1 h) and $n$th grid cell, respectively. Then, by assuming the airmass velocity is constant within
the time interval, the residence time in a grid cell ($T_{grid}$) was calculated by considering both the distance of each grid corner
($d_n$ and $d_{n+1}$) and the corresponding velocities ($V_n$ and $V_{n+1}$) using $d_n/V_n + d_{n+1}/V_{n+1}$. Based on the identified potential emission
region, APT was calculated only after the airmass passed through the potential emission region when precipitation occurred.
Figure 1c reveals the geographical distribution for the mean BC mass of identified potential emission regions, indicating that
this approach was appropriate because of good spatial coverage over East Asia, including East China, a major emission source
for BC.
**2.3 Transport efficiency (TE)**
The TE of BC is defined as the ratio of the BC and CO concentrations measured at the receptor site to that anticipated if
there was no wet removal during transport (i.e., APT is zero). Thus, the TE of the airmass was calculated by eq. (1),
$$TE = \frac{[\Delta BC / \Delta CO]_{APT>0}}{[\Delta BC / \Delta CO]_{APT=0}} \qquad (1)$$

where delta (Δ) indicates the difference between BC and CO concentrations and their baseline concentrations (Moteki et al.,
2012; Oshima et al., 2012; Kanaya et al., 2016). The baseline CO was estimated as a 14-day moving 5th percentile from the
observed CO mixing ratio, but the BC baseline was regarded as zero because the atmospheric lifetime of BC is known as
several days, which is much shorter than that of CO (1−2 months). $[\Delta BC/\Delta CO]_{APT=0}$ indicated the regional median value of
$\Delta BC/\Delta CO$ under dry conditions implying the original emission ratio. In our previous work, we successfully elucidated that
$[\Delta BC/\Delta CO]_{APT=0}$ depends on the regional characteristics of the energy consumption types (Kanaya et al., 2016; Choi et al.,
2020). The decrease of the ratio with APT, $[\Delta BC/\Delta CO]_{APT>0}$, was related to BC-specific removal due to wet scavenging
processes and thus the TE was effective indicator to investigate the wet removal process. Although TE is also affected by dry



deposition, but the effect of dry deposition could be negligible because dry deposition velocities were much lower than the
default setting in the global models (Choi et al., 2020).
**2.4 FLEXPART model**
To compare the wet removal rates between the model simulation and measured values, the FLEXPART v10.4 was used to
simulate BC wet scavenging over East Asia using the backward mode. Detailed information for the FLEXPART is readily
found in the literature (e.g., Stohl et al., 2005); thus, we only briefly describe the information here. The FLEXPART version
10.4 was the official version to allow turning on the wet scavenging module in the backward simulation mode
(https://www.flexpart.eu/downloads, obtained 10 October 2019). The equation and detailed description for the below- and in-
cloud scavenging scheme are explained in Pisso et al. (2019) and Grythe et al. (2017). The FLEXPART model was executed
with operational reanalysis meteorological data from the ECMWF ERA-Interim at a spatial resolution of 1°×1° with 60 full
vertical levels. Temporally, ECMWF has a resolution of 3 h, with 6 h analysis and 3 h forecast time steps. The period and
daily frequency of simulation were the same as those of the HYSPLIT model (past 72 h and four times, respectively). It should
be noted that chemistry and microphysics could not be resolved by the FLEXPART. The FLEXPART model, therefore, ignores
the aging process (from hydrophobic to hydrophilic state changes and size changes of BC) and assumes that all BC particles
are aged hydrophilic particles. The logarithmic size distribution of BC with a mean diameter of 0.16 μm and a standard
deviation of 1.84, in accordance with measurement in Japan, was used (Miyakawa et al., 2017). A total of $10^4$ particles were
randomly released at 500 m from each receptor site during 1 h when the measurement data were existed. To validate the wet
scavenging scheme in FLEXPART by comparison with the measured TE value, the wet scavenging coefficients for below- and
in-clouds were extracted from FLEXPART to calculate TE.
**3 Results**
**3.1 Overall variation of transport efficiency (TE)**
Figure 2 shows that measured $[\Delta BC/\Delta CO]_{APT=0}$ (left panel) and TE variations (right panel) depend on APT and the
measurement sites. The overall median $[\Delta BC/\Delta CO]_{APT=0}$ was 6.4 ng m$^{-3}$ ppb$^{-1}$, which converged from Baengnyeong (6.2 ng
m$^{-3}$ ppb$^{-1}$), Gosan (6.5 ng m$^{-3}$ ppb$^{-1}$) and Noto (6.7 ng m$^{-3}$ ppb$^{-1}$), indicating that TE is characterized with a regional
$[\Delta BC/\Delta CO]_{APT=0}$ per site. We divided APT into 9 range bins and applied exponential fitting equations to quantify the wet
removal process. Among $N_{APT>0}$ (total number of data points when APT > 0 mm), only the data point fraction in each bin to
$N_{APT>0} \geq 2\%$ was considered to secure the statistic. It should be noted that we found the relationship between TE and APT by
using the stretched exponential decay (SED) equation, $\exp(-A_1 \times APT^{A_2})$, instead of the widely used equation, $A-$
$B\times\log(APT)$, because the coefficients of determination ($R^2$) was improved up to 0.981 though TE values from three sites were
used (Table 1). This fitting equation is normally used to describe below-cloud scavenging, whereas wet removal of BC is
generally believed to be dominated by in-cloud rather than below-cloud processes because of the small size of BC-containing
particles. Therefore, the equations should contain both below- and in-cloud scavenging effects. The parameters $A_1$ (0.269 ±
0.039) and $A_2$ (0.385 ± 0.035) of the overall fitting were higher and lower, respectively, than the derived equation from the
Fukue site ($A_1$ = 0.109 and $A_2$ = 0.68) (Kanaya et al., 2016). It can be easily deduced that the wet removal effect at the three
sites was initially more effective than that at Fukue, but the wet removal effect at Fukue gradually accelerated as the APT
increased. In particular, the $A_2$ value is important for calculating the TE of BC for long-range transport, e.g., toward the Arctic





(Kanaya et al., 2016; Zhu et al., 2019), because $A_2$ determines the magnitude of the wet removal efficiency according to APT.
Thus, the newly obtained SED equation indicates that more BC will be transported to the Arctic region than previously reported.
The decreasing pattern of median TE for Baengnyeong did not closely follow the overall SED and had a much lower $R^2$
(0.77), indicating that the wet removal process at Baengnyeong could not simply be expressed by APT. In contrast, the $R^2$ of
Gosan and Noto were sufficiently high to represent the wet removal characteristics. The aging process due to different traveling
times might be one of the reasons. Because long-range transported BC has a larger core diameter than BC from local sources
(Lamb et al., 2018; Ueda et al., 2016), these larger BC cores are preferentially removed via the wet scavenging process (Moteki
et al., 2012). However, previous studies reported that the mass median diameter (MMD) of BC at Baengnyeong, Gosan, and
Noto in spring were 218, 196, and 200 nm (Oh et al., 2015; Ueda et al., 2016; Oh et al., 2014), respectively, indicating much
more aging compared with local emissions in Seoul, South Korea (180 nm) and Tokyo, Japan (163 nm) (Park et al., 2019;
Ohata et al., 2019). Moreover, there were no significant differences in the mean traveling times for the airmass (when APT >
0) arriving at the three sites (37.9, 39.0, and 37.8 h for Baengnyeong, Gosan, and Noto, respectively), indicating that the
difference in the level of the BC aging process might be negligible.
The difference in the wet removal rate among measurement sites could be partly explained by the difference in meteorology.
The monthly mean meteorological parameters indicated that Baengnyeong has characteristics of low precipitation (80.6 mm),
cloud cover (0.57), total column cloud water (0.06 kg m$^{-2}$), and high cloud bottom height (2.5 km) compared to other sites,
suggesting the lower exposure time to both below- and/or in-cloud condition during the transportation (Figure 3). In contrast,
the SED fittings for both Gosan and Noto showed similar ranges of high precipitation (127 and 174 mm), total cloud cover
(0.65 and 0.64), and total column cloud water (0.09 and 0.12 kg m$^{-2}$) but low cloud bottom height (1.9 and 2.0 km), respectively.
In addition, the different BC coating thicknesses according to the emission source and fuel types could also contribute to the
site difference of the wet removal rate, which will be further discussed in section 3.2.
Using the overall SED fitting equation, TE at 0.5 (TE=0.5) and $e$-folding (TE=1/$e$) could be reached when the APT values
were 11.7 and 30.2 mm, respectively (Table 1). Similar to the SED results, Baengnyeong needed much higher precipitation of
70.9 and 202 mm to reach TE=0.5 and TE=1/$e$, respectively, but the other sites showed lower APTs of 16.4 mm and 42.3 mm
for Gosan and 8.0 mm and 20.3 mm for Noto, respectively. Considering the annual mean precipitation at the three sites (1542
mm), it took 2.8 and 7.1 days to reach TE=0.5 and TE=1/$e$, respectively. Kanaya et al. (2016) reported a similar half-life and
shorter $e$-folding lifetime for BC at Fukue (2.3 ± 1.0 and 4.0 ± 1.0 days, respectively), calculated from the 15.0 ± 3.2 mm and
25.5± 6.1 mm of APT to reach TE=0.5 and TE=1/$e$, respectively, along with annual precipitation (2335 mm). This calculated
$e$-folding lifetime in East Asia was much shorter than 16.0 days for the global model (Grythe et al., 2017).
Based on a similar approach over the Yellow Sea using an aircraft-borne single particle soot photometer (SP2) during the
A-FORCE campaign (Oshima et al., 2012), attaining TE=0.5 required different magnitudes of APT depending on not only the
airmass origin but also the altitude. These authors also reported that the TE of northern China was higher than that of southern
China regardless of altitude. Therefore, in the next section, we will further investigate why the difference in halving or $e$-
folding lifetimes depends on region and season by analyzing the difference in the origin of airmasses and the seasonal variation
of BC emission sources.
**3.2 Regional and seasonal variations of the transport efficiency (TE)**
Figure 4 indicates the variation of TE depending on the potential source regions (hereafter regions) and seasons. The $R^2$ for



each source region was varied from 0.656 to 0.945 and was lower in East and North China and North Korea and higher in
other regions (Table 1). A similar tendency of $R^2$, TE=0.5 also showed different APTs, i.e., higher in East and North China and
lower in other regions. The regional difference in wet removal efficiency can partly be attributed to the following reasons.

First, the transport pathway of airmasses from East and North China could be less exposed to in-cloud scavenging than other

regions because the most of potential emission source in East and North China is located over 30°N (Figure 1c), which has
low cloud cover and water contents along with high cloud bottom heights (Figure 3). Although the amount of APT was similar
to other regions, it was mostly composed of below-cloud scavenging, therefore, the wet removal efficiency should be lower
than the dominant in-cloud scavenging region. Second, the difference in the coating thickness of BC particles, depending on
the emission sectors, could be a major factor causing the difference in the wet removal efficiency because thickly-coated BC
particles are much easier to remove by wet scavenging than less coated and/or freshly emitted BC (Miyakawa et al., 2017).
Typically, BC emitted from industrial regions, transport from diesel vehicles, and domestic sectors has characteristics of weakly,
moderate, and strongly coated BC, respectively (Han et al., 2019; Liu et al., 2019), based on insignificant differences in the
MMD of BC from those emission sectors (190 – 200 nm). This result coincided with the major emission sector of the REAS
emission inventory in East and North China and North Korea (~57.5% emitted from industrial sectors) compared to other sites
(12% − 39%). In contrast, Northeast China showed low APT for reaching TE=0.5 and TE=1/$e$ because the dominant BC
emission sector was residential sector (48.3%) which has a thickly coated characteristic. BC from South Korea and Japan
reached TE=0.5 and TE=1/$e$ with a small amount of APT because moderately coated BC was mostly emitted from the transport
sector (73.4%), mainly from diesel vehicles. It should be noted that the dominant emission sectors of industry (for East and
North China and North Korea) or transport sectors (South Korea and Japan) were also confirmed by the Emission Database
for Global Atmospheric Research (EDGAR) in 2010 and MIX in 2010 (Li et al., 2017; Crippa et al., 2018).

In case of seasonal variation in TE, the decreasing magnitude of TE was obviously emphasized in fall and winter, which

was much steeper than that in spring and summer (Figure 4b). This tendency was reflected in the effect of the residential sector,
which has thickly coated BC, which increased due to house heating as the temperature decreased. In contrast to winter, the
APT for reaching TE=0.5 in spring and summer was the highest among the seasons. This might be caused by the increasing
fraction of BC from the industrial sector in China while decreasing emissions from residential sectors (Kurokawa et al., 2013).
**3.3 Comparison of measured and FLEXPART-simulated TE**

In this section, by extracting the wet scavenging coefficients ($\Lambda$; s$^{-1}$) from the FLEXPART simulation, the difference in TE

between the measured and simulated values was investigated. The scavenging coefficient ($\Lambda$; s$^{-1}$) is defined as the rate of
aerosol washout and/or rainout due to the wet removal process. The TE value based on measurements and FLEXPART can be
expressed by multiplying each TE (1 – removal rate) of serial grid cells as in eq. (2),
$TE = (1-\eta_1)(1-\eta_2)\cdots(1-\eta_n)$                      (2)
where $\eta_n$ indicates the removal rate in the $n$th grid cell and is expressed as eq. (3),
$\eta = [1-\exp(-\Lambda \cdot t)] \cdot f_g$                       (3)
where $t$ and $f_g$ indicate the residence time and fraction for the subgrid in a grid cell, respectively. Because the precipitation is
not uniform in a single grid cell, $f_g$ accounts for the variability of precipitation in a grid cell in FLEXPART. $f_g$ is a function of





large-scale and convective precipitation, as described in Stohl et al. (2005). Although the grid resolution of the input
meteorological data for the HYSPLIT model (0.25°×0.25°) is much finer than that for FLEXPART (1°×1°), we assumed the
same potential emission region as the HYSPLIT model for calculating TE because there was no significant difference in the
airmass pathway between the two outputs.

The overall median value of measured TE was 0.72, and Baengnyeong showed the highest (0.88), followed by Gosan (0.70)
and Noto (0.68) due to reasons explained in the previous sections. In comparison, the overall median value of FLEXPART TE
(0.91) was much higher than the measured TE, indicating that the wet scavenging coefficients in the FLEXPART scheme were
significantly underestimated. Moreover, the difference in FLEXPART TE depending on the measurement sites (0.95 for
Baengnyeong, 0.94 for Gosan, and 0.87 for Noto) was not large as the measured TE, suggesting that the regional difference in
meteorological variables was relatively normalized and that the influence of other variables, which were not considered in the
wet scavenging scheme, might be excluded in the calculation. Meanwhile, it is difficult to capture the local variation from
coarse grid sizes, despite the airmass transport pathway between the two models being similar, because the key variables for
determining the wet scavenging coefficient (such as precipitation and cloud cover) could have a large local variability. In
addition, this approach still had a limitation in determining whether the overestimation of TE was resulting from the below- or
in-cloud scavenging processes. Nevertheless, with similar rationale, further comparison of measured and simulated scavenging
processes would provide information to better represent wet removal schemes.

### 3.4 Below-cloud scavenging efficiency ($\Lambda_{below}$)

From this section, we aimed to investigate the below- and in-cloud scavenging in detail by discriminating the representative
cases according to cloud information from the ERA5 pressure level data to overcome the limitation of the local variability of
meteorological input variables. By considering the vertical height of the airmass from the HYSPLIT model and cloud
information from ERA5, we successfully distinguished the dominant cases for below-cloud (no residence time within the cloud)
and in-cloud (no residence time below the cloud) cases when precipitation ≥ 0.01 mm hr$^{-1}$. The median TE and residence time
for only in-cloud cases (0.72 and ~7,200 h) were much lower and longer, respectively, than those for only below-cloud cases
(0.89 and ~5,100 h), indicating that most BC particles were effectively removed via the in-cloud scavenging process (Table 2).
In the case of below-cloud scavenging, the deviation of TE from unity could be simply converted to the scavenging coefficient
($\Lambda_{below}$) by considering the precipitation intensity, raindrop size, aerosol size, and residence time in a grid cell. Because many
studies have made an effort to parameterize $\Lambda_{below}$ using observation data and/or the theoretical calculation (Xu et al., 2017;
Wang et al., 2014b; Feng, 2007), we also parameterized this coefficient using a simplified method by following the scheme of
below-cloud scavenging in FLEXPART v10.4 (Laakso et al., 2003), which only considers the precipitation rate and aerosol
size. Assuming a BC size ~200 nm, TE for below-cloud can be expressed using equations (2) and (3) by substituting $\Lambda$ with
$\Lambda_{below}$, which depends only on the precipitation rate in the subgrid cell ($I_{total}$; the ratio of precipitation to $f_g$). Because $\Lambda_{below}$ can
be determined by constraining the proportion to the summation of $I_{total}$, hourly $\Lambda_{below}$ from the sequential grid cell in a single
case can easily be obtained by minimizing $\chi^2$, (TE$_{measured}$ − TE$_{calculated}$)$^2$ when $\chi^2 < 0.1$. This was conducted using an R function,
optimization (optim; https://stat.ethz.ch/R-manual/R-devel/library/stats/html/optim.html), included in the standard R package
"stats".

Figure 5a indicates the empirical cumulative density function for the measured $\Lambda_{below}$ from 869 cases. Although a substantial
fraction of $\Lambda_{below}$ was close to zero (or negative), the median $\Lambda_{below}$ was significantly different from zero and also positive
(7.9×10$^{-6}$ s$^{-1}$), with an interquartile range of −1.7×10$^{-5}$ s$^{-1}$ to 5.3×10$^{-5}$ s$^{-1}$. Negative $\Lambda_{below}$ values have been reported in previous



studies (Laakso et al., 2003; Pryor et al., 2016; Zikova and Zdimal, 2016); therefore, we assumed that these negative values
reflected the uncertainty in measurements and/or inclusion of BC, which might be continuously supplemented in airmasses.
As the threshold of $I_{total}$ increased from 0.01 (all cases) to 0.2 mm hr$^{-1}$ (median), $\Lambda_{below}$ values were increased by a factor of 2.5
to $2.0\times10^{-5}$ s$^{-1}$ ($-2.5\times10^{-5}$ s$^{-1}$ to $9.0\times10^{-5}$ s$^{-1}$). Using these obvious increasing tendencies of $\Lambda_{below}$ according to $I_{total}$, we
determined the empirical fitting equation by investigating the relationship between median $\Lambda_{below}$ and each bin of $I_{total}$. Figure
5b indicates $\Lambda_{below}$ as a function of $I_{total}$ by allocation to 11 logarithmic bins. As the estimated $I_{total}$ bins covered the $I_{total}$ ranges,
0.03 to 2.0 mm hr$^{-1}$ (5$^{th}$ percentile to 95$^{th}$ percentile), this exponential fitting equation ($A \times I_{total}{}^{B}$) could be representative for
below-cloud scavenging over East Asia. The constant A and exponent B with a 95% confidence interval were $2.0 \times 10^{-5}$ (1.9
$- 2.2 \times 10^{-5}$) and 0.54 (0.46 – 0.64), respectively. Instead of the SED equation shown in Figure 2, we chose the exponential
fitting equation because of its higher R$^2$ (0.973) compared to that from SED fitting (0.903), as well as being widely used in
previous studies.

Figure 6 shows the comparison of $\Lambda_{below}$ from reported values with this study by assuming that the BC size was

approximately 200 nm. To compare the measured $\Lambda_{below}$, we used mean fractional bias (MFB; $2\times[A - B]/[A + B]$), where $A$ and
$B$ denote $\Lambda_{below}$ of reported values and this study, respectively. Our newly measured $\Lambda_{below}$ values were located in the
intermediate range of reported $\Lambda_{below}$, and the mean deviations between measured and all reported values were relatively
constant with increasing $I_{total}$ because the mean absolute MFBs were slightly increased from 1.4 to 1.6. It should be noted that
$\Lambda_{below}$ from Laakso et al. (2003), which is the default scheme for below-cloud scavenging in the FLEXPART model version
10 or higher (Grythe et al., 2017), showed fairly good agreement with our $\Lambda_{below}$ among the reported values (mean absolute
MFBs was 0.68). MFB was positive at low $I_{total}$, but the tendency was opposite based on $I_{total} \sim 0.1$ mm hr$^{-1}$, suggesting that
$\Lambda_{below}$ might be converged within a similar range when we consider the range of $I_{total}$. Although $\Lambda_{below}$ from Laakso et al. (2003)
showed good agreement with our results, the median $\Lambda_{below}$ ($6.6 \times 10^{-6}$ s$^{-1}$) was overestimated compared to our estimation (4.0
$\times 10^{-6}$ s$^{-1}$), by a factor of 1.7 when we recalculated the only below-cloud cases. The MFBs from other schemes were too high
or low to declare reasonable results. For example, the $\Lambda_{below}$ of secondary ions in Beijing (Xu et al., 2017) had the highest MFB
(1.68), and although the diameter ranges were larger ($\sim 500$ nm) than those of BC, the effect of differences in diameter might
be negligible.
**3.5 In-cloud scavenging coefficient ($\Lambda_{in}$)**

Compared to $\Lambda_{below}$, the calculation of $\Lambda_{in}$ is much more complicated because many factors can influence the in-cloud

scavenging process, such as precipitation, total cloud cover (TCC), the specific cloud total water content (CTWC) and so on.
The detailed description for the complicated equation for $\Lambda_{in}$ in FLEXPART v10 is presented in Grythe et al. (2017), and the
equation for $\Lambda_{in}$ can be simply expressed as follows:
$$\Lambda_{in} = \frac{i_{cr} \cdot F_{nuc} \cdot I_{total} \cdot TCC}{CTWC \cdot f_g}$$ (4)
where $i_{cr}$ and $F_{nuc}$ are the cloud water replenishment factor (6.2; default value) and the nucleation efficiency, respectively. It
should be mentioned that $\Lambda_{in}$ was calculated by following the FLEXPART scheme using the ERA5 meteorological data
(0.25°×0.25°) instead of the FLEXPART simulation (1°×1°) to match the grid size of the input data with the HYSPLIT backward
trajectory. Among the 769 cases for in-cloud cases, equations (2) and (3) were also used to calculate TE for only in-cloud cases
by substituting $\Lambda$ with calculated $\Lambda_{in}$. Unlike the hourly measured $\Lambda_{below}$ calculated by optimization, the only overall median



$\Lambda_{in}$ ($\Lambda_{in}^*$) for in-cloud cases was calculated using equation (3) because $\Lambda_{in}$ cannot be constrained by a specific variable.
The FLEXPART $\Lambda_{in}^*$ ($7.28 \times 10^{-6}$ s$^{-1}$) was underestimated by 1 order of magnitude compared to our estimated $\Lambda_{in}^*$ ($8.06 \times$
$10^{-5}$ s$^{-1}$). When TE from FLEXPART for in-cloud cases (all cases) was recalculated by considering a ten (five) times higher
$\Lambda_{in}$, the median TE was 0.73 (0.79), which was much close to the measured TE (0.72). Although the grid size of input
meteorological data for two approaches did not match, the underestimation of the in-cloud scavenging scheme in FLEXPART
was confirmed. Grythe et al. (2017) reported an overestimation of observed BC (a factor of 1.68) due to the inaccurate emission
source rather than the underestimated in-cloud removal efficiencies. Although the effect of BC particle dispersion to adjacent
grid cells was neglected in our approach, the underestimation of in-cloud scavenging coefficients was obvious because the
accuracy of the emission inventory did not affect the estimated $\Lambda_{in}^*$. Looking more closely into the sites, the FLEXPART $\Lambda_{in}^*$
at Noto was remarkably underestimated by 1 order of magnitude, followed by Gosan (~90%) and Baengnyeong (~43%),
similar to the order of the wet removal efficiency. It should be noted that the coefficient of variation (CV; standard deviation
divided by the mean) of FLEXPART $\Lambda_{in}^*$ was much lower (0.23) than the measured $\Lambda_{in}^*$ (0.78), indicating that FLEXPART
$\Lambda_{in}^*$ did not accurately represent the actual regional difference in the real world. Among the input meteorological variables in
equation (4), the CV of $I_{total}$ was the highest as 0.22, which was similar to the CV of FLEXPART $\Lambda_{in}^*$, followed by CTWC
(0.08), $f_g$ (0.03), and TCC (0.02), suggesting that the difference in FLEXPART $\Lambda_{in}^*$ could be partially explained by $I_{total}$ rather
than other variables. Among the meteorological variables that were not considered in equation (4), the convective available
potential energy (CAPE), which is well known as an indicator of vertical instability (Mori et al., 2014), had the highest CV of

0.31.

We employed an artificial neural network (ANN) method to compare the importance of CAPE with other considered input
meteorological variables for determining the hourly $\Lambda_{in}$, not $\Lambda_{in}^*$. We applied a stricter selection for in-cloud cases that only
when in-cloud scavenging occurred less than three times (i.e., three cells) in a single case, regardless of the number of below-
cloud occurrences. Because the effect of below-cloud scavenging was successfully excluded from the TE using the derived
equation for $\Lambda_{below}$ in the previous section, the $\Lambda_{in}$ in less than three in-cloud cases can also be calculated by optimization based
on the remaining TE. We applied a threshold of three cases here because the number of data (230 cases) was sufficient to
conduct statistical analysis, while the optimization uncertainty could be reduced to its minimum. The ANN model was trained
using the six meteorological variables (CAPE, CTWC, $f_g$, $F_{nuc}$, $I_{total}$, and TCC), and all variables were normalized by the
minimum and maximum of each variable ($[x\text{-min}(x)]/[\max(x)\text{-min}(x)]$). To determine optimal node numbers in the hidden
layer, we applied a 'caret' package of the R function that contain several sets of machine learning modes and validation tools
(https://cran.r-project.org/web/packages/caret/caret.pdf) and adopted a method from the 'neuralnet' package that is fit for a
multi-hidden layer. By varying the 'size' (node number) from 5 to 20 and using $k$-fold cross validation, the selected cases were
randomly divided 3:1 into training (172 data points) and validation data (58 data points). Garson's algorithm in the
"NeuralNetTools" package was used to identify the relative importance of six input variables in the final neural network
(Garson, 1991). The model's performance was assessed in these independent validation data by calculating the root mean
squared error. The optimal number of nodes in the hidden layer was 12 (Figure 7a).
Figure 7b shows the relative importance of input variables for calculating $\Lambda_{in}$ using Garson's algorithm. The most important
input variable was CAPE, with a value of 35%, followed by CTWC, $I_{total}$, and so on, confirming that CAPE should be
considered in the $\Lambda_{in}$ calculation. Typically, enhancing wet removal by convective clouds successfully reduces the aloft BC
concentration in the free troposphere (Koch et al., 2009). Therefore, convective process is important in tropical regions but has





a slightly lower impact at mid-latitudes (Luo et al., 2019; Grythe et al., 2017; Xu et al., 2019). Moreover, previous studies
pointed out convective scavenging to be a key parameter in determining the BC concentration in model simulation (Lund et
al., 2017; Xu et al., 2019) and the role of wet removal by convective clouds might be significant when most airmasses travel
above the planetary boundary layer. Unfortunately, the current version of FLEXPART does not implement convective
scavenging (Philipp and Seibert, 2018), which could be a plausible reason for the underestimation of FLEXPART $\Lambda_{in}$. Although
the relative importance of each variable cannot be parameterized to calculate $\Lambda_{in}$, this approach highlights that CAPE is one
of the key factors for determining $\Lambda_{in}$ over East Asia. In the future, more information might be required to evaluate the in-
cloud scavenging scheme using Weather Research and Forecasting (WRF)-FLEXPART at a higher resolution in further studies,
since a 0.25º grid size is still not sufficient to reproduce convective clouds (typically 10 km or less).

## 4 Conclusions

The wet removal rates and scavenging coefficients for BC were investigated by the term of $\Delta BC/\Delta CO$ ratios from long-
term, best-effort observations at three remote sites in East Asia (Baengnyeong and Gosan in South Korea and Noto in Japan).
Combined with backward trajectories covering the past 72 h, accumulated precipitation along trajectories (APT), and transport
efficiency (TE; $[\Delta BC/\Delta CO]_{APT>0}/[\Delta BC/\Delta CO]_{APT=0}$), the assessment of BC wet removal efficiency was conducted as an aspect
of the pathway of trajectories, including the successful identification of below- and in-cloud cases. The overall wet removal
rates as a function of APT, the half-life and $e$-folding lifetime were similar to those of previous studies but showed large
regional differences depending on the measurement sites. The difference in the wet removal rate, depending on the
measurement site, can be explained by the different meteorological conditions, such as the precipitation rate, cloud cover, total
column cloud water, and cloud bottom height. Moreover, the difference in regional or seasonal wet removal rates could be
explained by the different coating thicknesses according to the BC emission sources (thin- and thick-coated BC from the
industrial and residential sectors, respectively) because the thick-coated BC particles are preferentially removed due to cloud
processes. By discriminating below- and in-cloud dominant cases according to cloud vertical information from ERA5 pressure
level data, scavenging coefficients for below-cloud ($\Lambda_{below}$) and in-cloud ($\Lambda_{in}^{*}$) were simply converted from the measured TE
values. The $\Lambda_{below}$ from the FLEXPART scheme was overestimated by a factor of 1.7 compared to the measured $\Lambda_{below}$, although
the measured $\Lambda_{below}$ showed good agreement with the below-cloud scheme in FLEXPART among the reported scavenging
coefficients. In contrast to $\Lambda_{below}$, FLEXPART $\Lambda_{in}^{*}$ was highly underestimated by 1 order of magnitude compared to measured
$\Lambda_{in}^{*}$, suggesting that the current in-cloud scavenging scheme did not represent regional variability. By diagnosing the relative
importance of the input variables using the artificial neuron network (ANN) method, we found that the convective available
potential energy (CAPE), which is an indicator of vertical instability, should be considered to improve the in-cloud scavenging
scheme because convective scavenging could be regarded as a key parameter for determining the accurate BC concentration
in a model. This study could contribute not only to improving the below-cloud scavenging scheme implemented in a model,
especially FLEXPART, but also to providing evidence for complementary in-cloud scavenging schemes by considering the
convective scavenging process. For the first time, these results suggest a novel and straightforward approach to evaluating the
wet scavenging scheme in various models and to enhancing the understanding of BC behavior by excluding the effect of
inaccurate emission inventory.



**Author contributions.**

YC and YK designed the study and prepared the paper, with contributions from all co-authors. YC, MT, and CZ optimized the FLEXPART model and revised the paper. YC simulated the FLEXPART model and conducted analyses. SMP was responsible for measurements at Baengnyeong. AM and YS conducted measurements at Noto, and SWK contributed to ground observations and quality control at Gosan. XP and IP contributed to the data analysis. All co-authors provided professional comments to improve the paper.

**Competing interests.**

The authors declare that they have no conflicts of interest.

**Code/Data availabilty.**

The observational data set for BC and CO are available upon request to the corresponding author.

**Acknowledgments.**

The authors thank NOAA ARL and ECMWF for providing the HYSPLIT model and ERA5 meteorological data.

**Financial support.**

This research has been supported by the Environment Research and Technology Development Fund of the Ministry of the Environment, Japan (grant no. 2-1803).





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





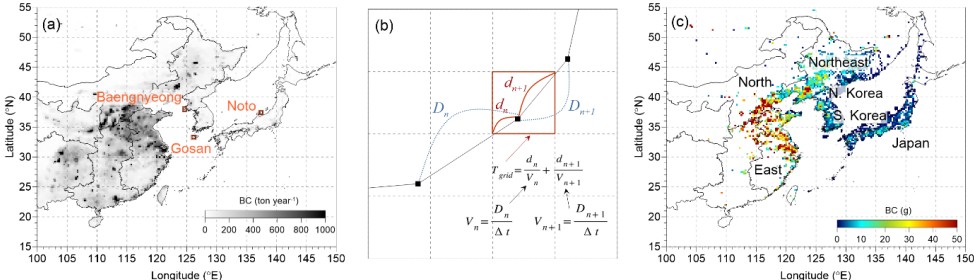

**Figure 1.** (a) The location of three measurement sites (Baengnyeong, Gosan, and Noto) and the black carbon (BC) emission rate (ton year$^{-1}$) over East Asia from the Regional Emission inventory in ASia (REAS) version 2.1 (Kurokawa et al., 2013). (b) Illustration of residence time calculated based on the HYSPLIT backward trajectory that passed over a single grid cell (see details in the manuscript). (c) The spatial distribution of the mean BC mass in the potential emission region, which is the highest BC mass grid of each trajectory. The BC mass was obtained by multiplying (a) the emission rates and (b) the residence time.

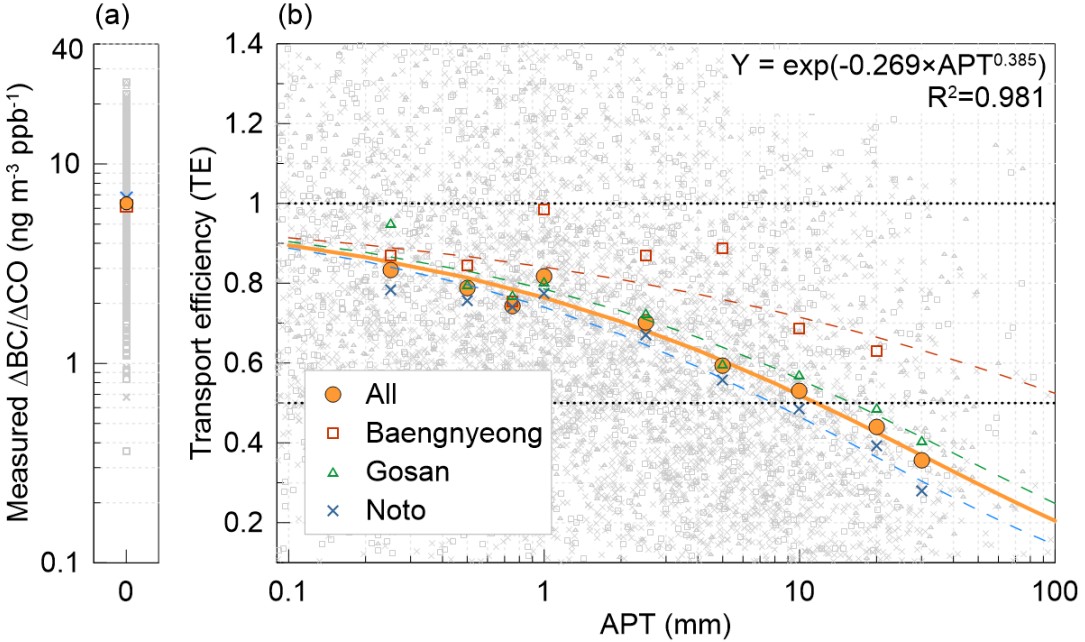

**Figure 2.** Measured ΔBC/ΔCO ratios when accumulated precipitation along trajectory (APT) was zero (left panel) and transport efficiency (TE) variation as a function of APT (right panel) depending on the different sites and overall cases. All data (gray with different symbols) and 9 bins sorted by APT (different colored symbols) are shown. The horizontal dotted lines indicate TE at 0.5 and 1, respectively.



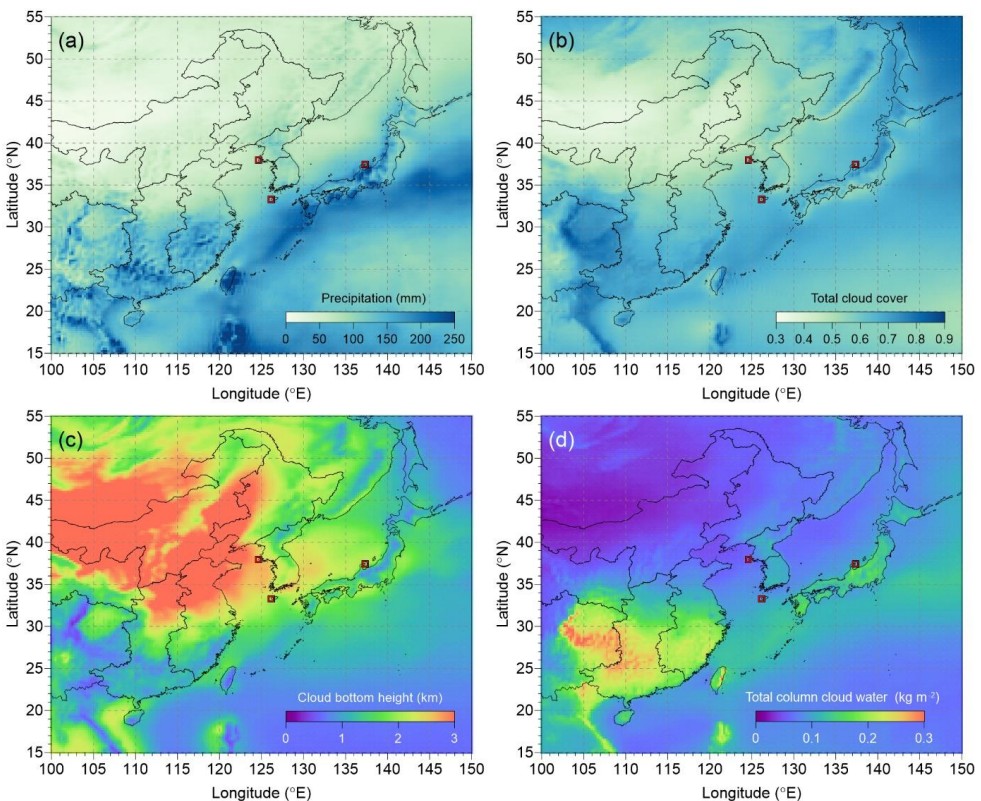

**Figure 3.** Monthly mean meteorological fields over East Asia from 2010 to 2016 derived from the European Centre for Medium-Range Weather Forecasts (ECMWF) ERA5 monthly averaged data at single levels, (a) precipitation (mm), (b) total cloud cover, (c) cloud bottom height (km), and (d) total column cloud total water (ice and liquid).

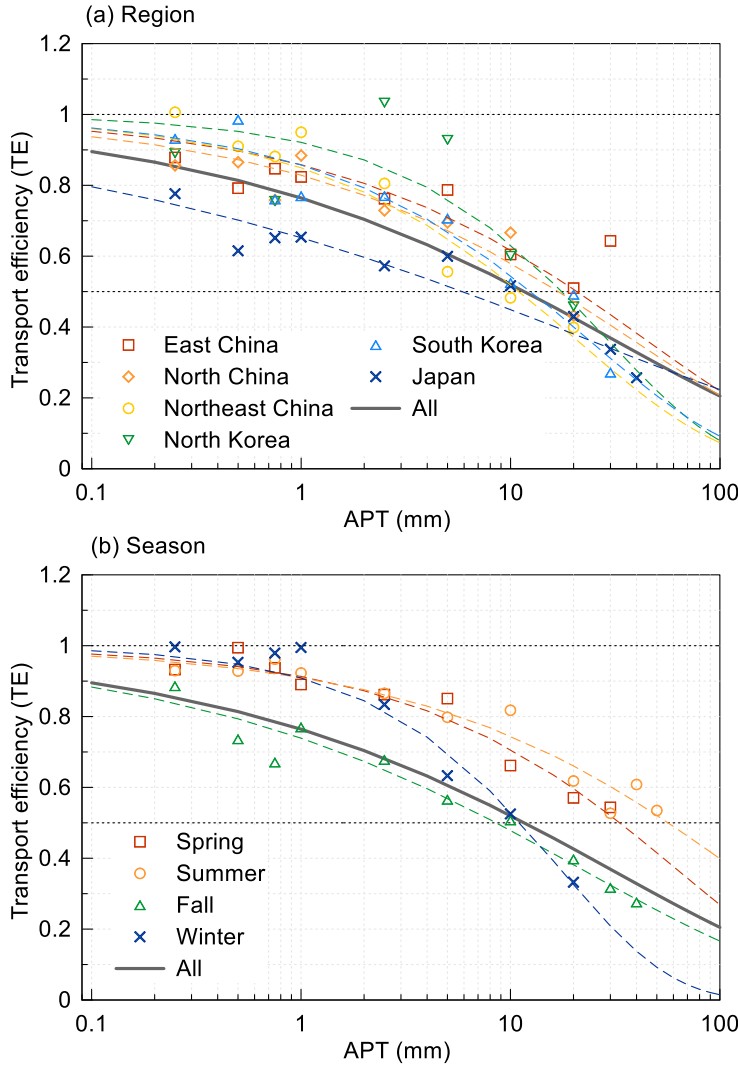

**Figure 4.** Same as Figure 2 except for (a) regional and (b) seasonal variation of TE according to APT. Each colored symbol and dashed line indicate the different regions and seasons and fitting lines according to stretched exponential decay (SED). The thick gray line depicts the overall fitting line. The horizontal dotted lines indicate TE at 0.5 and 1, respectively.



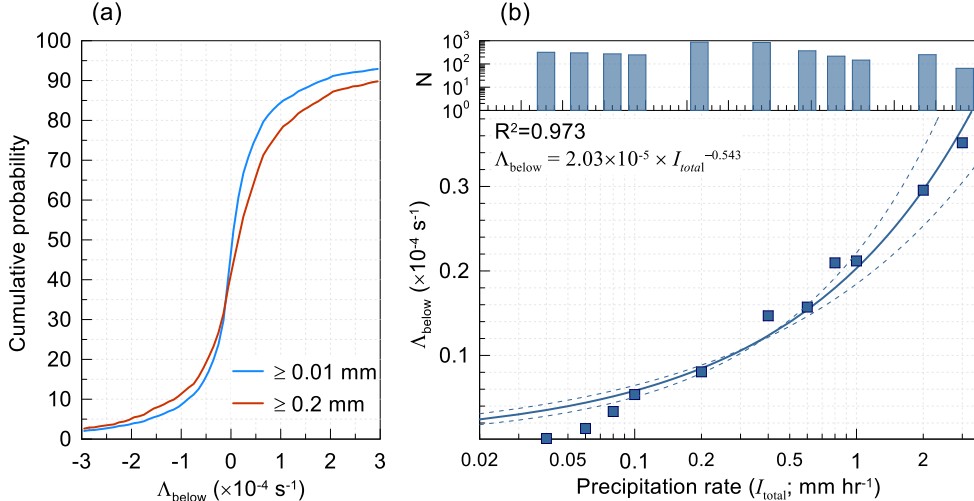

**Figure 5.** (a) Empirical cumulative distribution plot of measured below-cloud scavenging coefficients ($\Lambda_{below}$; s$^{-1}$) depending on the precipitation rate ($\geq 0.01$ and $\geq 0.2$ mm hr$^{-1}$). (b) Median measured $\Lambda_{below}$ as a function of the precipitation intensity (mm hr$^{-1}$) of 11 bins. The dashed line indicates the fit from the equation. The upper panel of (b) shows the number of hourly data points for each bin for $I_{total}$.





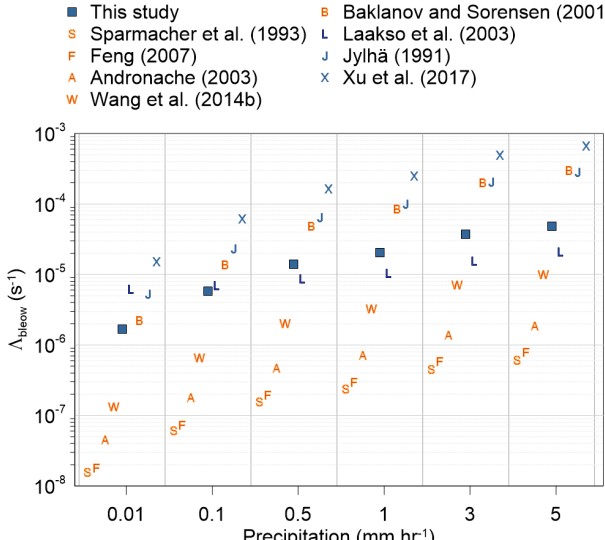

**Figure 6.** Variation of reported and measured below-cloud scavenging coefficients ($\Lambda_{below}$; $s^{-1}$) depending on the precipitation intensity (mm $hr^{-1}$). Orange and blue symbols depict the $\Lambda_{below}$ equation based on theoretical calculation and observation data, respectively. The diameter of BC was assumed to be approximately 200 nm in the calculation.



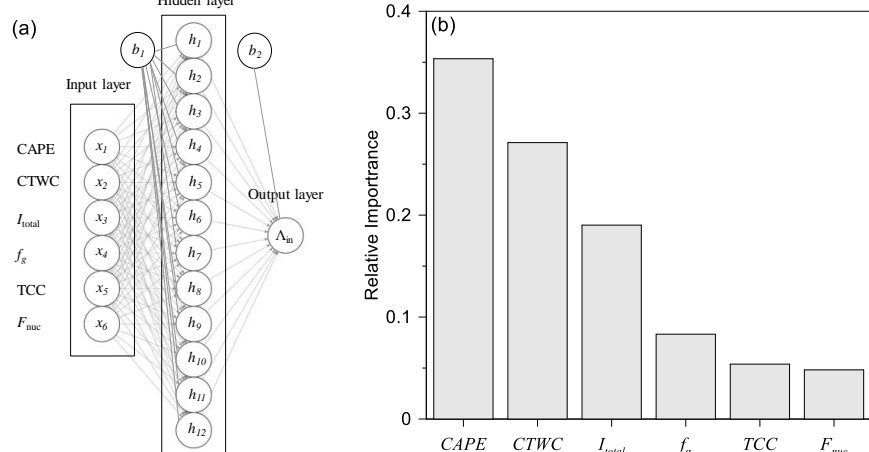

**Figure 7.** (a) Schematic of an artificial neuron network (ANN) model with 12 nodes of a single hidden layer. (b) The relative importance of six input meteorological variables used for calculating in-cloud scavenging coefficients in the FLEXPART model (except for CAPE) using Garson's algorithm implemented in the 'NeuralNetTools' package in R. CAPE, CTWC, $I_{total}$, $f_g$, TCC, and $F_{nuc}$ represent the convective available potential energy, specific cloud total water content, precipitation rate, fraction of a subgrid in a grid cell (see manuscript for details), total cloud cover, and nucleation efficiency, respectively.





**Table 1.** Summary of the relationship between transport efficiency (TE) and accumulated precipitation along trajectory (APT) in Figures 2 and 4.

| | Fitting parameters[a] | | $R^2$ | APT (mm) | | Number of data points | | Days | | Annual |
|---|---|---|---|---|---|---|---|---|---|---|
| | $A_1$ | $A_2$ | | TE=0.5 | TE=1/$e$ | $N_{APT=0}$ | $N_{APT>0}$[b] | TE=0.5 | TE=1/$e$ | Precipitation (mm) |
| All | 0.269 ± 0.039 | 0.385 ± 0.035 | 0.981 | 11.7 | 30.2 | 3,565 | 6,611 | 2.8 | 7.1 | 1542.3 |
| | | | | | | | | | | |
| Site | | | | | | | | | | |
| Baengnyeong | 0.156 ± 0.117 | 0.350 ± 0.146 | 0.773 | 70.9 | 201.9 | 1,732 | 1,522 | 35.5 | 101.2 | 728.3 |
| Gosan | 0.235 ± 0.047 | 0.386 ± 0.047 | 0.964 | 16.4 | 42.3 | 705 | 1,090 | 4.9 | 12.5 | 1233.3 |
| Noto | 0.306 ± 0.052 | 0.393 ± 0.036 | 0.985 | 8.0 | 20.3 | 1,128 | 4,057 | 1.1 | 2.8 | 2665.3 |
| | | | | | | | | | | |
| Region | | | | | | | | | | |
| East | 0.153 ± 0.099 | 0.498 ± 0.183 | 0.866 | 20.7 | 43.3 | 439 | 704 | | | |
| North | 0.188 ± 0.090 | 0.462 ± 0.175 | 0.897 | 16.9 | 37.3 | 518 | 495 | | | |
| Northeast | 0.163 ± 0.084 | 0.603 ± 0.166 | 0.945 | 11.0 | 20.3 | 1,237 | 2,175 | | | |
| N. Korea | 0.082 ± 0.414 | 0.745 ± 0.813 | 0.656 | 17.5 | 28.7 | 216 | 393 | | | |
| S. Korea | 0.154 ± 0.110 | 0.596 ± 0.188 | 0.922 | 12.5 | 23.2 | 325 | 680 | | | |
| Japan | 0.428 ± 0.117 | 0.272 ± 0.089 | 0.925 | 5.9 | 22.6 | 687 | 1,789 | | | |
| | | | | | | | | | | |
| Season | | | | | | | | | | |
| Spring | 0.122 ± 0.045 | 0.506 ± 0.111 | 0.957 | 31.2 | 64.5 | 1,285 | 1,366 | | | |
| Summer | 0.143 ± 0.107 | 0.362 ± 0.182 | 0.780 | 77.3 | 212.6 | 497 | 1,685 | | | |
| Fall | 0.288 ± 0.055 | 0.397 ± 0.057 | 0.972 | 9.1 | 23.0 | 767 | 1,606 | | | |
| Winter | 0.070 ± 0.048 | 0.905 ± 0.192 | 0.964 | 12.5 | 18.7 | 1,016 | 1,986 | | | |

[a] $TE = \exp(-A_1 \times APT^{A_2})$

[b] The number of satisfactory data points in each bin relative to total $N_{APT>0} \geq 2\%$





**Table 2.** Summaries of the transport efficiency (TE) and scavenging coefficients for selected (a) below- and (b) in-cloud cases based on ERA5 hourly data of pressure levels from ECMWF.

| Cases | Median | Interquartile range (25th percentile – 75th percentile) |
|---|---|---|
| (a) Below cloud ($N_{case}$ = 831) | | |
| TE | 0.89 | [0.61 – 1.27] |
| Estimated $\Lambda_{below}$ (s$^{-1}$) | $4.01\times10^{-6}$ | [$2.70\times10^{-6}$ – $6.33\times10^{-6}$] |
| FLEXPART $\Lambda_{below}$ (s$^{-1}$) | $6.63\times10^{-6}$ | [$6.38\times10^{-6}$ – $7.08\times10^{-6}$] |
| | | |
| (b) In-cloud ($N_{case}$ = 769) | | |
| TE | 0.72 | [0.43 – 1.06] |
| Estimated $\Lambda_{in}$* (s$^{-1}$) [a] | $8.06\times10^{-5}$ | - |
| FLEXPART $\Lambda_{in}$* (s$^{-1}$) [a] | $7.28\times10^{-6}$ | - |

[a] Overall median value