# Peer review of "Investigation of the wet removal rate of black carbon in East Asia: validation of a below- and in-cloud wet removal scheme in FLEXPART v10.4"

_Atmospheric Chemistry and Physics, 2020_

## Referee Comment (RC1) · Anonymous Referee #1 · 10 Jun 2020

General comments:

This paper introduces a method to access the wet removal rate of BC in East Asia based on long-term measurements, in the aspect of the air mass back trajectories. The authorship made effort to obtain the overall wet removal rates of BC as a function of accumulated precipitation along trajectories, the half-life and e-folding lifetime. Depending on the measurement sites, the wet removal rates of BC showed large regional differences, and various reasons are explored. Further, they diagnosed the scavenging coefficients of the below- and in-cloud scavenging scheme implemented in the FLEXible PARTicle (FLEXPART) Lagrangian transport model with the obtained wet removal rates of BC, and suggested that underestimation of wet scavenging coefficients in the model simulation. Finally, they evaluated the relative importance of various factors in the in-cloud scavenging process, and indicated that the convective available potential energy should be considered to better represent the regional difference of BC wet scavenging over East Asia.

The topic of the manuscript is well suited for publication in ACP. The long-term dataset are generally applicable, whereas some discussions are lack of persuasion. I suggest more effort should be put into the presentation of the results before publication. My major concern is about the preset for the calculation and the reasons for the regional difference of wet removal efficiency.

(1) The authors used 500 m as a starting altitude and 72h back trajectories were calculated. Is it an arbitrary selection? How does this affect the final assessment of wet removal?

(2) The authors attributed the regional difference in wet removal efficiency to the difference in the coating thickness of BC particles. In the discussion section, they consider that depending on the emission sectors, the coating thickness of BC particles could be a major factor causing the difference in the wet removal efficiency. I think such explanation is hard to believe. The freshly emitted BC particles has transported for a long distant before scavenged. How could the freshly emitted BC particles affect their coating thickness before scavenged? Actually, there are many published paper showing factors that drive the ageing of BC, which should be included in the discussion.

Specific comments:

1. Introduction: "Specifically, the in-cloud process is more efficient and complicated than the below-cloud process because the nucleation removal of aerosol particles within clouds is thought to account for more than 50% of the aerosol particle mass removal from the atmosphere globally" I wonder if there are any scavenging efficiency

data for BC alone, since this paper mainly focus on the wet removal of BC.

2. Introduction: "Wet deposition is still challenging to predict BC concentration in the atmosphere due to the difficulties of accurate evaluation of wet removal." It would be better to include more explanation on why it is challenging to represent wet deposition, which tightly links to the discussion section of this paper.

3. Introduction: It would be better to simply explain "emission rates and deposition terms".

4. Experimental section: "when the airmass altitude was lower than 2.5 km...". Is there any explanation for this?

5. Line 199-: Is there any correlation between wet removal of BC and meteorological parameters?

Minor:

Line 65: "stat"?

Line 73: "significant"?

Line 136: "good spatial coverage"?

Line 149: "thus the TE was an effective indicator".

Line 193: "However"?
* * *

---

## Referee Comment (RC2) · Anonymous Referee #2 · 20 Jul 2020

General comments:

The paper addresses a topic of scientific relevance which is within the scope of Atmospheric Chemistry and Physics. They present a method to evaluate the wet removal rate of black carbon (BC) with the LPDM Flexpart based on long-term measurements over East Asia. The authors used back-trajectories with Hysplit to identify the source region of the air masses from 3 stations over East Asia and to determine the accumulated precipitation along the trajectories. With this information, they calculated the Transport Efficiency (TE) of black carbon from the measurements and compared the TE

from measurements with the results of the backward modelling with Flexpart v10.4 to assess the simulated wet removal rate in transport modelling. Additionally, the authors further distinguish their evaluation between below cloud and in-cloud wet removal by diagnosing the scavenging coefficients. They show that the wet removal of BC from the different measurement sites have significant differences and discuss various reasons. According to the scavenging from below and in-cloud, they found that Flexpart under-estimates the in-cloud scavenging and overestimates the below-cloud scavenging. By using a neural network, the authors investigated the importance of several dependencies. They found that CAPE is the parameter with the most substantial influence on in-cloud scavenging and suggest to include this parameter in the future.

However, even though the methods and discussions provide valuable assets to the community, they are not easy to understand and clear in the way it was written. I had to re-read multiple sections to identify what was done and which values from which data sets were compared or used. In my opinion, this could be improved by some additional definitions and distinctions. For example, the authors should clearly distinguish between measured data, determined/calculated data and simulated data. For example, I often got confused by the mentioning of Flexpart scheme, since this was sometimes the simulated data and sometimes the algorithm for deposition calculation.

I suggest major revisions as outlined in the comments below and addressing the specific comments before publishing the manuscript.

Major comments:

1. How did you select the simulation setup? Why only 72h of backward runs and why starting with a height of 500m for the release location? There must have be some investigation or thought on this. What effect does it have on the results?

2. Why do you use two different models with a different data set each? This causes a lot of differences and uncertainties in the results. You mention that you did not find large differences in the pathways between Hysplit with ERA5 and Flexpart with ERA-

[Figure]

Interim. But there are still differences due to the different physical parameterizations and spatial and temporal resolutions. Wouldn't it be more accurate to use Flexpart for the trajectory calculations also and therefore use the same data set? I know that ERA5 model level data were not easily available for all users in the past, but it is now. Therefore, it would be a substantial improvement to use the same data set, namely ERA5, for all simulations. I am aware that this is probably not possible to achieve within this review process, but the authors should discuss this and provide more details about possible uncertainties.

3. It is not recommended to use all four analysis times (0,6,12 and 18 UTC) per day and combine them with forecast fields to achieve 3-hourly temporal resolution. This causes unnecessary inconsistencies between 5h and 7h as well as 17h and 19h. The recommendation is to use 0 and 12 UTC and fill the times in between with forecast fields. I also thought that ERA-Interim on model levels were only available at 0,6,12 and 18UTC for public users, which gives me the indication that the access method was as a member-state user? Is this correct? Then you should have had access to ERA5 data all the time anyway.

4. After going through the manuscript I had a hard time to distinguish which data set and scheme/formula was used to calculate TE or scavenging coefficients. I would highly suggest to go through section 3.4 and 3.5 (below and in-cloud scavenging) again and try to be more clear in the description and distinguishing of where a scavenging coefficient comes from. Maybe by giving it different subscriptions.

Specific comments:

1. p.1 l.29: You mention diagnosing the scavenging coefficients from Flexpart. I thought that Flexpart defines the coefficients upfront in a species file. Therefore, I don't understand why the coefficients need to be diagnosed. Could you explain please?

2. p.2 l. 58: . . . because TE has been proven . . .. ; could you provide evidence?

3. p.2 l. 65: what is meant by "mixing stats" ?

4. p2. l.68-71: This sentence is hard to understand, especially the part with "...polluted areas scavenging in cloud were more...".

5. p.2 l.73: . . . could be a useful parameter . . . ; I thought it is a useful parameter, why could?

6. p.2 l.74: You mention that you adopt APT. You adopt it from where and how?

7. p.3 l.82: What are the administrative districts? Could you provide a plot?

8. p3. l.85: Again, you estimate the scavenging coefficients from FLEXPART? Why? I sense that I might miss or misunderstand something.

9. p.3 l.93: What does "intensive" in this context mean? Do you really mean intensive?

10. p.3 l.98: "The measurement periods were mainly in the early 2010s . . . "; Do you mean that they start in the early 2010s ?

11. p.3 l.99: Since Figure S1 is in the supplement, you might want to add a note on that.

12. p.3 l.101: "well-validated" ; What is well-validated? There should be a criterion for this.

13. p.4 l.118: Why mentioning GDAS?

14. p.4 l.119: Do the pressure levels correspond to ERA5 or GDAS?

15. p.4 l.121: Did you disaggregate the precipitation fields as they are done for Flexpart simulations? For better comparison to Flexpart results.

16. p.4 l.124: What are the main BC regions? How is "main BC region" defined?

17. p.4 l.125: Why couldn't the precipitation not be overestimated?

18. p.5 l.150: What do the global models have to do with this study?

19. p5. l.155: shouldn't there be a reference to the Flexpart v10.4 paper?

20. p.5 l.159: ERA-Interim is not an operational reanalysis. ERA-Interim was suspended and ERA5 is now operational!

21. p.5 l.159-160: You should rewrite to "60 model levels" since vertical levels could also be pressure levels or others.

22. p. 5 l.160: "ECMWF has a resolution of 3h..." this is wrong. The ERA-Interim data set has this resolution, but not ECMWF in general.

23. p.5 l.168: what do you mean by "extracted" ? How do you calculate the TE with Flexpart data.

24. p.5 l.174: Could you define the bins somewhere? This should be done according to be able to reproduce the results.

25. p.5 l.178: What was $R^2$ before it was improved?

26. p.5 l.183: This Fukue site comes out of nowhere and it is not clear where it is located.

27. p.6 l.187: I don't understand how the new SED indicates the transport to the Artic, please explain further.

28. p.6 l.214: what global model?

29. p.7 l.224: " A similar tendency of $R^2$, TE=0.5 also showed .. " ; I don't understand this formulation. Do you mean $R^2$ and TE ?

30. p.8 l.258: wasn't this described in Grythe et al. 2017 ?

31. p.8 l.258 – 261: Its not only the grid resolution but the whole model physics is different apart from the differences between Hysplit and Flexpart. Additionally, regional/local pattern of precipitation and clouds are totally different especially because Flexpart uses disaggregated precipitation while it seems that Hysplit and ERA5 data used in this

study weren't disaggregated. How does this reflect in the results? And why did you chose 1°x1° for ERA-Interim instead of the 0.75°x0.75° resolution ERA-Interim was stored on?

32. p.8 l. 274.: In this section, I got confused by the values from measured vs ERA5 vs calculated vs reported vs Flexpart. Did you calculate the TE with the Flexpart scheme from ERA5 data? Then, what did you use from Flexpart simulation results?

33. p.8 l.301: again, could you please define the bins somewhere? (reproducabilty)

34. p.8 l.307: what are reported values?

35. p.8 l.319: could you give a reference for your statement of " the effect of differences in diameter might be negligible" please?

36. p.10 l.333: the Flexpart scavenging coefficient is taken from the simulations with ERA-Interim data and the estimated coefficient is from the measurement data in combination with the scheme from Flexpart and ERA5 data? Is this correct?

37. p10. l.362: what would be the effect if it would be 4:1 or 2:1?

38. Table 2: Does this mean Flexpart does not correspond to the ERA-Interim simulation results but to the calculated values with ERA5 data and the Flexpart scheme? If not, reformulate please.

Technical corrections:

p.3 l.83: Afterward → Afterwards p.5 l.150: remove "but" p.5 l.150: negligible → neglected p.5 l.153: I would suggest to exchange the order of "model simulation" and "measured values" p.5 l.166: were existed → were available p.7 l.223: was varied → varied p.7 l.230: " than the dominat in-cloud . . ." → than in the dominant ... p.7 l.249: simulation → simulations

---

## Author Comment (AC1) · 28 Aug 2020

**Response to Referee #1**

*General Comments: This paper introduces a method to access the wet removal rate of BC in East Asia based on long-term measurements, in the aspect of the air mass back trajectories. The authorship made effort to obtain the overall wet removal rates of BC as a function of accumulated precipitation along trajectories, the half-life and e-folding lifetime. Depending on the measurement sites, the wet removal rates of BC showed large regional differences, and various reasons are explored. Further, they diagnosed the scavenging coefficients of the below- and in-cloud scavenging scheme implemented in the FLEXible PARTicle (FLEXPART) Lagrangian transport model with the obtained wet removal rates of BC, and suggested that underestimation of wet scavenging coefficients in the model simulation. Finally, they evaluated the relative importance of various factors in the in-cloud scavenging process, and indicated that the convective available potential energy should be considered to better represent the regional difference of BC wet scavenging over East Asia. The topic of the manuscript is well suited for publication in ACP. The long-term dataset are generally applicable, whereas some discussions are lack of persuasion. I suggest more effort should be put into the presentation of the results before publication. My major concern is about the preset for the calculation and the reasons for the regional difference of wet removal efficiency.*

Response: We thank the reviewer for carefully reviewing the manuscript and providing valuable comments. We also acknowledge your valuable comments and suggestions that greatly helped to improve the manuscript. The following are our responses to your specific comments. For convenience, your comments are italicized and numbered. The line (L) numbers in the responses correspond to those in the revised manuscript. The changes in the revised manuscript are underlined in the responses as necessary, and are indicated as 'tracked changes' in the manuscript.

1. *The authors used 500 m as a starting altitude and 72h back trajectories were calculated. Is it an arbitrary selection? How does this affect the final assessment of wet removal?*

   We replaced the past 72 h backward trajectory, which can represent the wet deposition effects, to the past 120 h by considering the BC lifetime (~5 d) and including dry deposition effects; however, the results are exactly the same as in the original manuscript because identified potential emission source regions are consistent with the original manuscript. The difference in the starting altitude (500 m *vs.* 1000 m) did not impact our results; i.e., the ranges of the TE for sites, regions and seasons used in Table 1 and below- and in-cloud cases in Table 2 were similar to the original results (Sect. S1 in the Supplement). A detailed explanation of the uncertainty due to the selection of different starting altitudes was addressed as follows:

   "To identify the airmass origin region, 5 d ( 120 h) backward trajectories were calculated four times a day (00, 06, 12, 18 UTC) using the Hybrid Single Particle Lagrangian Integrated Trajectory (HYSPLIT) Model version 4 (Draxler et al., 2018). The starting altitude was 500 m above ground level (AGL). The past 120 h of backward simulation time was selected by considering the lifetime of BC (~5 d; Lund et al., 2017, 2018; Park et al., 2005). It should be noted that the different starting altitude (500 m *vs.* 1000m) did not impact on our results (Sect. S1 in the Supplement)." (L140−144)

   "Our main results, including the TE, $\Lambda_{below}$, and $\Lambda_{in}$, could be influenced by selecting (1) different starting altitudes of the backward trajectories and (2) different altitude criteria for identifying the potential emission region.

   First, to investigate the uncertainty caused by different starting altitudes of the backward trajectories, we analyzed the Welch's *t*-test for APT derived from starting altitudes of 500 m and 1000 m. The APT between the two datasets did not show a significant difference (3%) ($p \geq 0.1$). Depending on the site, the TE showed a significant difference ($p < 0.05$) at Gosan only at a relatively small value of −4.2%. In the case of regional TE, Northeast China and South Korea were significantly different ($p < 0.01$), with original values up to −15%; however, the corresponding APT for achieving TE=0.5 and TE=1/*e*

only decreased by −6% and −2%, respectively. The regional wet removal efficiency was more apparent, such as more or less APT needed to attain TE=0.5 and TE=1/e in low-efficiency regions (East and North China) and high-efficiency regions (South Korea and Japan), respectively. For the high starting altitude, i.e., 1000 m, the airmass had a higher chance of being exposed to in-cloud scavenging resulting in a much lower TE for in-cloud scavenging (−3%). Otherwise, the TE for below-cloud scavenging cases was increased by 7% because of a reduced chance to expose washout effects (Table S1). Because of the variations in the TE for below- and in-cloud scavenging cases, the calculated median $\Lambda_{below}$ and $\Lambda_{in}$ converged within a similar range as the original results. It should be noted that the median measured $\Lambda_{below}$ was slightly higher than the calculated $\Lambda_{below}$ according to FLEXPART, which is opposite the original results. The small difference could be ignored when considering the insufficient sample number for below-cloud cases at a starting altitude of 1000 m." (Sect. S1 in the Supplement)

**Table S1.** Same as Table 2 except for the different backward trajectory starting altitudes (1000 m)

| Cases | Median | Interquartile range (25th percentile – 75th percentile) |
|---|---|---|
| (a) Below cloud ($N_{case}$ = 262) | | |
| TE | 0.95 | [0.65 – 1.28] |
| Measured $\Lambda_{below}$ (s$^{-1}$) | $8.85\times10^{-6}$ | [$6.57\times10^{-6}$ − $1.46\times10^{-5}$] |
| Calculated $\Lambda_{below}$ (s$^{-1}$) [a] | $7.49\times10^{-6}$ | [$6.83\times10^{-6}$ − $8.42\times10^{-6}$] |
| (b) In-cloud ($N_{case}$ = 953) | | |
| TE | 0.70 | [0.46 – 1.02] |
| Measured $\Lambda_{in}$* (s$^{-1}$) [b] | $7.67\times10^{-5}$ | - |
| Calculated $\Lambda_{in}$* (s$^{-1}$) [a,b] | $8.01\times10^{-6}$ | - |

[a] Calculated by FLEXPART scheme
[b] Overall median value

2. *The authors attributed the regional difference in wet removal efficiency to the difference in the coating thickness of BC particles. In the discussion section, they consider that depending on the emission sectors, the coating thickness of BC particles could be a major factor causing the difference in the wet removal efficiency. I think such explanation is hard to believe. The freshly emitted BC particles has transported for a long distant before scavenged. How could the freshly emitted BC particles affect their coating thickness before scavenged? Actually, there are many published paper showing factors that drive the ageing of BC, which should be included in the discussion.*

We agree with the reviewer's opinion that the BC aging process is most important when considering the predominance of in-cloud scavenging. Therefore, we added the description of the BC aging process and frequency of below- and in-cloud scavenging conditions as the most plausible reasons causing regional and seasonal differences in wet scavenging efficiency. The explanation of difference in coating thickness of BC upon emission was removed because of the lack of evidence supporting our hypothesis.

[revised manuscript text omitted]
" I wonder if there are any scavenging efficiency data for BC alone, since this paper mainly focus on the wet removal of BC.*

We replaced the sentences to focus on BC particles as follows:

"Wet deposition of BC, whose contribution to total removal is 79% (Textor et al., 2006), is still challenging to predict BC concentrations in the atmosphere due to the difficulties of accurate evaluation of wet removal (Emerson et al., 2018; Bond et al., 2013; Lee et al., 2013). Specifically, the in-cloud process is more efficient and complicated than the below-cloud process because the nucleation removal of aerosol particles within clouds is thought to account for  46 ± 50% of the BC particle mass removal from the atmosphere globally, although this is dependent on the selected global model (Grythe et al., 2017; Textor et al., 2006)." (L61−66)

2. *Introduction: "Wet deposition is still challenging to predict BC concentration in the atmosphere due to the difficulties of accurate evaluation of wet removal." It would be better to include more explanation on why it is challenging to represent wet deposition, which tightly links to the discussion section of this paper.*

We added a detailed description of the reasons for the difficulties in accurately evaluating the wet removal of BC as follows:

"This can partly be attributed to the following three reasons: (1) inaccurate bottom-up emission inventory, (2) the complexity of BC hygroscopicity, and (3) an imprecise dry/wet deposition scheme. First, when estimating the impact of BC using global models, the results usually contain large uncertainties in BC emissions (Cooke and Wilson, 1996; Chung and Seinfeld, 2002; Stier et al., 2007) because BC is mainly contributed by scattered emission sources. Therefore, the uncertainty of BC emission rates is large compared to other species (e.g., $SO_2$, $NO_x$, and $CO_2$) whose emissions are dominated by large sources (Kurokawa et al., 2013; Zheng et al., 2018). Without appropriate constraints on the emissions, removal cannot be well quantified. Second, although BC itself is hydrophobic immediately after emission, it is subsequently converted to possessing hydrophilic properties through the aging process, in which water-soluble compounds coat BC. during atmospheric transportation (Moteki et al., 2007; Matsui et al., 2018), and finally acts as cloud condensation nuclei (Kuwata et al., 2007; Bond et al., 2013). Such conversion depends on the initial state of the BC along with atmospheric conditions (presence of other particles and gases) and it has high spatial and temporal variabilities (Vignati et al., 2010)." (L49−59)

"However, there is insufficient in-field detailed observations to explain and quantify the interactions between BC and cloud particles at the microscale, which hinders a better understanding of the physical processes (Ding et al., 2019)." (L66−68)

3. *Introduction: It would be better to simply explain "emission rates and deposition terms".*

We provided additional detail regarding the 'emission rates and deposition terms' as follows:

"Although some previous studies have investigated wet scavenging schemes in models (Grythe et al., 2017; Croft et al., 2010), those results without well-constrained emission rates contain large ambiguity when assessing the wet deposition term (Vignati et al., 2010)  For the first time, the emission and deposition terms are distinctly separated in this study by introducing TE and using backward simulations, thus allowing for the wet scavenging scheme to be evaluated more accurately because backward simulations do not account for the emission rate." (L92−98)

4. *Experimental section: "when the airmass altitude was lower than 2.5 km…". Is there any explanation for this?*

We added a discussion of the uncertainty in the criteria for selecting the altitude as follows:

"We checked the uncertainty arising from selecting different criteria for altitude (1.5 km), but there was no significant difference in the results (Sect. S1 in the Supplement)." (L171−172)

"Second, we also checked the difference in wet scavenging efficiency, which can be caused by applying 1.5 km (instead of 2.5 km) as a threshold to determine the potential emission region. The identified six administrative districts for potential emission regions at an altitude of 1.5 km were same as those at an altitude of 2.5 km. The median traveling time from potential source regions to receptor sites was decreased from 38 h to 25 h when precipitation occurred because the individual potential source region was closer to the receptor site because the selection altitude was decreased. However, the difference in traveling time did not significantly influence our final results because the TE for below- and in-cloud cases only decreased by 1% and 6% and the measured $\Lambda_{below}$ and $\Lambda_{in}$ were consistent with the original results within ±54% (Table S2). From these results, we confirmed the representativeness of our regional and seasonal wet removal efficiency analysis." (Sect. S1 in the the Supplement)

**Table S2.** Same as Table 2 except for the different altitude criteria (1.5 km) for identifying potential emission source regions.

| Cases | Median | Interquartile range (25th percentile – 75th percentile) |
|---|---|---|
| (a) Below cloud ($N_{case}$=436) | | |
| TE | 0.88 | [0.60 – 1.24] |
| Measured $\Lambda_{below}$ (s$^{-1}$) | 6.17×10$^{-6}$ | [2.55×10$^{-6}$ – 1.39×10$^{-5}$] |
| Calculated $\Lambda_{below}$ (s$^{-1}$) [a] | 7.52×10$^{-6}$ | [6.88×10$^{-6}$ – 8.50×10$^{-6}$] |
| | | |
| (b) In-cloud ($N_{case}$=282) | | |
| TE | 0.68 | [0.44 – 1.03] |
| Measured $\Lambda_{in}$* (s$^{-1}$) [b] | 9.39×10$^{-5}$ | - |
| Calculated $\Lambda_{in}$* (s$^{-1}$) [a,b] | 8.15×10$^{-6}$ | - |

[a] Calculated by FLEXPART scheme
[b] Overall median value

5. *Line 199-: Is there any correlation between wet removal of BC and meteorological parameters?*

The median TE as a function of the site showed a good correlation with the precipitation rate and total column cloud water (R≥0.94) and moderate correlation with cloud cover and cloud bottom heights (R≥0.52) because only three data records were available. Thus, we did not add the R value between two variables in this qualitative analysis section because a detailed analysis was conducted in the following sections according to the back trajectories along with their meteorological conditions.

*Minor:*

1. *Line 65: "stat"?*

   We revised it to 'states.' (L81)

2. *Line 73: "significant"?*

   We changed this term to 'considerable.'

   "Recently, numerous fine mode particles, including BC, from polluted areas scavenging in clouds were more pronounced in East Asia, not only at a local scale but also at a large regional scale (Liu et al., 2018), because high aerosol loading conditions are usually associated with considerable  cloud cover, which results in a higher frequency of wet scavenging (Eck et al., 2018).." (L84−87)

3. *Line 136: "good spatial coverage"?*

   We replaced 'good spatial coverage' with a detailed description.

   "Figure 1c reveals the geographical distribution for the mean BC mass of identified potential emission regions, indicating that this approach was appropriate because  the potential emission regions were uniformly distributed over East Asia, including East China, a major emission source for BC." (L167−169)

4. *Line 149: "thus the TE was an effective indicator".*

   We revised it as follows:

   "... thus the TE  is an effective indicator ..." (L184)

5. *Line 193: "However"?*

[revised manuscript text omitted]

**S1 Uncertainty in the transport efficiency (TE) and below- and in-cloud scavenging coefficients**

Our main results, including the TE, $\Lambda_{below}$, and $\Lambda_{in}$, could be influenced by selecting (1) different starting altitudes of the backward trajectories and (2) different altitude criteria for identifying the potential emission region.

First, to investigate the uncertainty caused by different starting altitudes of the backward trajectories, we analyzed the Welch's $t$-test for APT derived from starting altitudes of 500 m and 1000 m. The APT between the two datasets did not show a significant difference (3%) ($p \geq 0.1$). Depending on the site, the TE showed a significant difference ($p < 0.05$) at Gosan only at a relatively small value of −4.2%. In the case of regional TE, Northeast China and South Korea were significantly different ($p < 0.01$), with original values up to −15%; however, the corresponding APT for achieving TE=0.5 and TE=1/$e$ only decreased by −6% and −2%, respectively. The regional wet removal efficiency was more apparent, such as more or less APT needed to attain TE=0.5 and TE=1/e in low-efficiency regions (East and North China) and high-efficiency regions (South Korea and Japan), respectively. For the high starting altitude, i.e., 1000 m, the airmass had a higher chance of being exposed to in-cloud scavenging resulting in a much lower TE for in-cloud scavenging (−3%). Otherwise, the TE for below-cloud scavenging cases was increased by 7% because of a reduced chance to expose washout effects (Table S1). Because of the variations in the TE for below- and in-cloud scavenging cases, the calculated median $\Lambda_{below}$ and $\Lambda_{in}$ converged within a similar range as the original results. It should be noted that the median measured $\Lambda_{below}$ was slightly higher than the calculated $\Lambda_{below}$ according to FLEXPART, which is opposite the original results. The small difference could be ignored when considering the insufficient sample number for below-cloud cases at a starting altitude of 1000 m.

Second, we also checked the difference in wet scavenging efficiency, which can be caused by applying 1.5 km (instead of 2.5 km) as a threshold to determine the potential emission region. The identified six administrative districts for potential emission regions at an altitude of 1.5 km were same as those at an altitude of 2.5 km. The median traveling time from potential source regions to receptor sites was decreased from 38 h to 25 h when precipitation occurred because the individual potential source region was closer to the receptor site because the selection altitude was decreased. However, the difference in traveling time did not significantly influence our final results because the TE for below- and in-cloud cases only decreased by 1% and 6% and the measured $\Lambda_{below}$ and $\Lambda_{in}$ were consistent with the original results within ±54% (Table S2). From these results, we confirmed the representativeness of our regional and seasonal wet removal efficiency analysis.

**S2 Difference in airmass pathways and accumulated precipitation along trajectory (APT) between HYSPLIT and FLEXPART**

We investigated the uncertainty in the airmass pathway and APT between HYSPLIT model using ERA5 and FLEXPART using ERA-Interim during study periods at three sites. It should be noted that the trajectory of FLEXPART was selected as the center of the main grid (1°×1°) according to the highest residence time in the same time interval and then compared with HYSPLIT results by calculating the distance between two hourly endpoints. Thus, differences of less than ~100 km can be regarded as a good agreement when considering the grid resolution of FLEXPART. The difference in distance increased as the traveling time was increased. However, the median traveling time of airmasses, including APT=0 case, was 31 h, which showed a difference in distance of ~100 km. When the traveling time was expanded up to the 75%ile of the traveling time (50 h), the difference in distance was close to ~200 km. Although the difference in distance at 72 h traveling time was high, 72 h traveling time cases was so rare that we could neglect the impacts on our results. In total, the median difference in distance was ~47 km, suggesting good agreement between the two datasets. In addition, the difference in accumulated backward-trajectory endpoints was much smaller because random errors in the single calculations can be diminished by increasing the number of calculations (Gebhart et al., 2005; Jeong et al., 2017).

Figure S2 presents the cumulative probability of APT from HYSPLIT and FLEXPART. Although the airmass pathway showed insignificant differences between the two models, the median APT of FLEXPART (1.2 mm) was two times higher than that of HYSPLIT (0.63 mm), indicating a higher bias of the FLEXPART APT. This result can be caused by the difference in meteorological input data and the treatment of precipitation fields, homogeneous precipitation in a single grid cell ($0.25° \times 0.25°$) in HYSPLIT and disaggregated precipitation induced by interpolating in time and space in FLEXPART (Hittmeir et al., 2018). The higher bias in the FLEXPART APT contributed to increasing the magnitude of the underestimation of FLEXPART TE when assuming the same APT from HYSPLIT model, indicating an insignificant impact on the results.

Gebhart, K.A., Schichtel, B.A., Barna, M.G.: Directional biases in back trajectories caused by model and input data. J. Air Waste Manag. 55, 1649-1662, 2005.

Jeong, U., Kim, J., Lee, H., Lee, Y.G.: Assessing the effect of long-range pollutant transportation on air quality in Seoul using the conditional potential source contribution function method. Atmos. Environ., 150, 33-44, 2017.

Hittmeir, S., Philipp, A., and Seibert, P.: A conservative reconstruction scheme for the interpolation of extensive quantities in the Lagrangian particle dispersion model FLEXPART. Geosci. Model Dev., 11, 2503-2523, https://doi.org/10.5194/gmd-11-2503-2018, 2018.

[Figure]

**Figure S1.** Time series of the ΔBC/ΔCO ratio and accumulated precipitation along trajectory (APT) during the measurement periods in (a) Baengnyeong (1 Jan 2010–31 Dec 2016), (b) Gosan (1 May 2012–30 Apr 2015), and (c) Noto (1 Jan 2011–31 Dec 2016). The square symbols with solid lines indicate monthly concentrations. The red shaded region in the Baengnyeong figure indicates periods of data missing from 2011 to 2012 due to the absence of CO data.

[Figure]

**Figure S2.** Cumulative probability plot of the APT from HYSPLIT (blue) and FLEXPART (red) during the study periods at the three sites. The dashed black line indicated a cumulative probability at 0.5 (median).

**Table S1.** Same as Table 2 except for the different backward trajectory starting altitudes (1000 m)

| Cases | Median | Interquartile range (25th percentile – 75th percentile) |
|---|---|---|
| (a) Below cloud ($N_{case}$ = 262) | | |
| TE | 0.95 | [0.65 – 1.28] |
| Measured $\Lambda_{below}$ ($s^{-1}$) | $8.85 \times 10^{-6}$ | [$6.57 \times 10^{-6}$ – $1.46 \times 10^{-5}$] |
| Calculated $\Lambda_{below}$ ($s^{-1}$) [a] | $7.49 \times 10^{-6}$ | [$6.83 \times 10^{-6}$ – $8.42 \times 10^{-6}$] |
| | | |
| (b) In-cloud ($N_{case}$ = 953) | | |
| TE | 0.70 | [0.46 – 1.02] |
| Measured $\Lambda_{in}$* ($s^{-1}$) [b] | $7.67 \times 10^{-5}$ | - |
| Calculated $\Lambda_{in}$* ($s^{-1}$) [a,b] | $8.01 \times 10^{-6}$ | - |

[a] Calculated using FLEXPART scheme
[b] Overall median value

**Table S2.** Same as Table 2 except for the different altitude criteria (1.5 km) for identifying potential emission source regions.

| Cases | Median | Interquartile range (25th percentile – 75th percentile) |
|---|---|---|
| (a) Below cloud ($N_{case}$ = 436) | | |
| TE | 0.88 | [0.60 – 1.24] |
| Measured $\Lambda_{below}$ ($s^{-1}$) | $6.17 \times 10^{-6}$ | [$2.55 \times 10^{-6}$ – $1.39 \times 10^{-5}$] |
| Calculated $\Lambda_{below}$ ($s^{-1}$) [a] | $7.52 \times 10^{-6}$ | [$6.88 \times 10^{-6}$ – $8.50 \times 10^{-6}$] |
| | | |
| (b) In-cloud ($N_{case}$ = 282) | | |
| TE | 0.68 | [0.44 – 1.03] |
| Measured $\Lambda_{in}$* ($s^{-1}$) [b] | $9.39 \times 10^{-5}$ | - |
| Calculated $\Lambda_{in}$* ($s^{-1}$) [a,b] | $8.15 \times 10^{-6}$ | - |

[a] Calculated using FLEXPART scheme
[b] Overall median value

---

## Author Comment (AC2) · 28 Aug 2020

**Response to Reviewer #2**

*General comments:*

*The paper addresses a topic of scientific relevance which is within the scope of Atmospheric Chemistry and Physics. They present a method to evaluate the wet removal rate of black carbon (BC) with the LPDM Flexpart based on long-term measurements over East Asia. The authors used back-trajectories with Hysplit to identify the source region of the air masses from 3 stations over East Asia and to determine the accumulated precipitation along the trajectories. With this information, they calculated the Transport Efficiency (TE) of black carbon from the measurements and compared the TE from measurements with the results of the backward modelling with Flexpart v10.4 to assess the simulated wet removal rate in transport modelling. Additionally, the authors further distinguish their evaluation between below cloud and in-cloud wet removal by diagnosing the scavenging coefficients. They show that the wet removal of BC from the different measurement sites have significant differences and discuss various reasons. According to the scavenging from below and in-cloud, they found that Flexpart underestimates the in-cloud scavenging and overestimates the below-cloud scavenging. By using a neural network, the authors investigated the importance of several dependencies. They found that CAPE is the parameter with the most substantial influence on in-cloud scavenging and suggest to include this parameter in the future.*

Response: We thank the reviewer for carefully reviewing the manuscript and providing valuable comments. We also acknowledge your valuable comments and suggestions that greatly helped to improve the manuscript. The following are our responses to your specific comments. For convenience, your comments are italicized and numbered. The line (L) numbers in the responses correspond to those in the revised manuscript. The changes in the revised manuscript are underlined in the responses as necessary and indicated using 'tracked changes' in the manuscript.

*However, even though the methods and discussions provide valuable assets to the community, they are not easy to understand and clear in the way it was written. I had to re-read multiple sections to identify what was done and which values from which data sets were compared or used. In my opinion, this could be improved by some additional definitions and distinctions. For example, the authors should clearly distinguish between measured data, determined/calculated data and simulated data. For example, I often got confused by the mentioning of Flexpart scheme, since this was sometimes the simulated data and sometimes the algorithm for deposition calculation. I suggest major revisions as outlined in the comments below and addressing the specific comments before publishing the manuscript.*

We tried to clarify the terminology through the entire manuscript. Only the measured TE was compared with the simulated TE from FLEXPART, and the measured below- and in-cloud scavenging coefficients were compared with those calculated according to the FLEXPART scheme. Therefore, we used different terminology when addressing the FLEXPART value: 'simulated' for the output from FLEXPART and 'calculated' for following the FLEXPART scheme. In light of the comments, we have now unified the terminology from 'estimated' to 'measured' when discussing the outputs derived from the measured TE. To prevent misunderstanding, we added the following statements:

"Note that the simulated TE from FLEXPART (FLEXPART TE) was only used for comparing with the measured TE." (L206−207)

"From this section, we aimed to investigate the below- and in-cloud scavenging in detail by discriminating the representative cases according to cloud information from the ERA5 pressure level data with HYSPLIT backward trajectory to overcome the limitation of the local variability of meteorological input variables." (L344−346)

"..., we compared our measured scavenging coefficients with those calculated according to FLEXPART scheme (not simulated)." (L349−350)

"It should be mentioned that $\Lambda_{in}$ was also calculated by following the FLEXPART scheme using the ERA5 meteorological data (0.25°×0.25°) with HYSPLIT backward trajectory instead of the FLEXPART simulation (1°×1°) to reflect the local variability of meteorological variables ." (L400−402)

"The  calculated $\Lambda_{in}$* (7.28×10⁻⁶ s⁻¹) from FLEXPART scheme (hereafter calculated $\Lambda_{in}$*) was underestimated by 1 order of magnitude compared to our  measured $\Lambda_{in}$* (8.06×10⁻⁵ s⁻¹)." (L406−407)

*Major comments:*

1. *How did you select the simulation setup? Why only 72h of backward runs and why starting with a height of 500m for the release location? There must have be some investigation or thought on this. What effect does it have on the results?*

We replaced the past 72 h backward trajectory, which can represent the wet deposition effects, to the past 120 h by considering the BC lifetime (~5 d) and including dry deposition effects; however, the results are exactly the same as in the original manuscript because identified potential emission source regions are consistent with the original manuscript. The difference in the starting altitude (500 m *vs.* 1000 m) did not impact our results; i.e., the ranges of the TE for sites, regions and seasons used in Table 1 and below- and in-cloud cases in Table 2 were similar to the original results (Sect. S1 in the Supplement). A detailed explanation of the uncertainty due to the selection of different starting altitudes was addressed as follows:

"To identify the airmass origin region, 5 d ( 120 h) backward trajectories were calculated four times a day (00, 06, 12, 18 UTC) using the Hybrid Single Particle Lagrangian Integrated Trajectory (HYSPLIT) Model version 4 (Draxler et al., 2018). The starting altitude was 500 m above ground level (AGL). The past 120 h of backward simulation time was selected by considering the lifetime of BC (~5 d; Lund et al., 2017, 2018; Park et al., 2005). It should be noted that the different starting altitude (500 m *vs.* 1000m) did not impact on our results (Sect. S1 in the Supplement)." (L140−144)

"Our main results, including the TE, $\Lambda_{below}$, and $\Lambda_{in}$, could be influenced by selecting (1) different starting altitudes of the backward trajectories and (2) different altitude criteria for identifying the potential emission region.

First, to investigate the uncertainty caused by different starting altitudes of the backward trajectories, we analyzed the Welch's *t*-test for APT derived from starting altitudes of 500 m and 1000 m. The APT between the two datasets did not show a significant difference (3%) ($p \geq 0.1$). Depending on the site, the TE showed a significant difference ($p < 0.05$) at Gosan only at a relatively small value of −4.2%. In the case of regional TE, Northeast China and South Korea were significantly different ($p < 0.01$), with original values up to −15%; however, the corresponding APT for achieving TE=0.5 and TE=1/*e* only decreased by −6% and −2%, respectively. The regional wet removal efficiency was more apparent, such as more or less APT needed to attain TE=0.5 and TE=1/e in low-efficiency regions (East and North China) and high-efficiency regions (South Korea and Japan), respectively. For the high starting altitude, i.e., 1000 m, the airmass had a higher chance of being exposed to in-cloud scavenging resulting in a much lower TE for in-cloud scavenging (−3%). Otherwise, the TE for below-cloud scavenging cases was increased by 7% because of a reduced chance to expose washout effects (Table S1). Because of the variations in the TE for below- and in-cloud scavenging cases, the calculated median $\Lambda_{below}$ and $\Lambda_{in}$ converged within a similar range as the original results. It should be noted that the median measured $\Lambda_{below}$ was slightly higher than the calculated $\Lambda_{below}$ according to FLEXPART, which is opposite the original results. The small difference could be ignored when considering the insufficient sample number for below-cloud cases at a starting altitude of 1000 m." (Sect. S1 in the Supplement)

**Table S1.** Same as Table 2 except for the different backward trajectory starting altitudes (1000 m)

| Cases | Median | Interquartile range (25$^{th}$ percentile – 75$^{th}$ percentile) |
|---|---|---|
| **(a) Below cloud ($N_{case}$ = 262)** | | |
| TE | 0.95 | [0.65 – 1.28] |
| Measured $\Lambda_{below}$ (s$^{-1}$) | $8.85\times10^{-6}$ | [$6.57\times10^{-6}$ – $1.46\times10^{-5}$] |
| Calculated $\Lambda_{below}$ (s$^{-1}$) [a] | $7.49\times10^{-6}$ | [$6.83\times10^{-6}$ – $8.42\times10^{-6}$] |
| | | |
| **(b) In-cloud ($N_{case}$ = 953)** | | |
| TE | 0.70 | [0.46 – 1.02] |
| Measured $\Lambda_{in}$* (s$^{-1}$) [b] | $7.67\times10^{-5}$ | - |
| Calculated $\Lambda_{in}$* (s$^{-1}$) [a,b] | $8.01\times10^{-6}$ | - |

[a] Calculated by the FLEXPART scheme
[b] Overall median value

2. *Why do you use two different models with a different data set each? This causes a lot of differences and uncertainties in the results. You mention that you did not find large differences in the pathways between Hysplit with ERA5 and Flexpart with ERA Interim. But there are still differences due to the different physical parameterizations and spatial and temporal resolutions. Wouldn't it be more accurate to use Flexpart for the trajectory calculations also and therefore use the same data set? I know that ERA5 model level data were not easily available for all users in the past, but it is now. Therefore, it would be a substantial improvement to use the same data set, namely ERA5, for all simulations. I am aware that this is probably not possible to achieve within this review process, but the authors should discuss this and provide more details about possible uncertainties.*

We agree with the reviewer's opinion that the consistency of the FLEXPART simulation with ERA5 could be strengthened to demonstrate the robustness of our results compared with the use of only the results from the HYSPLIT model. However, we do not have authorization to access ERA5 through 'flex_extract' because we are a 'public user' who can access the ERA-Interim. However, the ERA5 for HYSPLIT model is open access.

Although we used different meteorological fields (ERA5 for HYSPLIT and ERA-Interim for FLEXPART), the calculated below- and in-cloud scavenging coefficients were derived using HYSPLIT model with ERA5 vertical meteorological information according to the FLEXPART scheme. Only the TE were compared and verified with that from FLEXPART with ERA-Interim. Although the detailed meteorological parameters were different according to their temporal (1 h *vs.* 6 h) and spatial (0.25° *vs.* 1°) resolutions, Hoffmann et al. (2019) reported that the particle positions (latitude, longitude, and altitude) along a 10-day forward trajectory calculated with both ERA-Interim and ERA5 showed good agreement. We also demonstrated an insignificant difference in airmass pathways between HYSPLIT and FLEXPART, and the following was added to the manuscript:

"Despite the difference in the input meteorological fields between HYSPLIT and FLEXPART, the difference in airmass pathways and APT between two datasets can be neglected (Hoffmann et al., 2019; Sect. S2 in the Supplement)." (L207−L209)

"We investigated the uncertainty in the airmass pathway and APT between HYSPLIT model using ERA5 and FLEXPART using ERA-Interim during study periods at three sites. It should be noted that the trajectory of FLEXPART was selected as the center of the main grid (1°×1°) according to the highest residence time in the same time interval and then compared with HYSPLIT results by calculating the distance between two hourly endpoints. Thus, differences of less than ~100 km can be regarded as a good agreement when considering the grid resolution of FLEXPART. The difference in distance increased as the traveling time was increased. However, the median traveling time of airmasses, including APT=0 case, was 31 h, which showed a difference in distance of ~100 km. When the traveling time was expanded up to the 75%ile of the traveling time (50 h), the difference in distance was close to ~200 km. Although the difference in distance at 72 h traveling time was high, 72 h traveling time cases was so rare that we could neglect the impacts on our results. In total, the median difference in distance was ~47 km, suggesting good agreement between the two datasets. In addition, the difference in accumulated backward-trajectory endpoints was much smaller because random errors in the single calculations can be diminished by increasing the number of calculations (Gebhart et al., 2005; Jeong et al., 2017)." (Sect. S2 in the Supplement)

3. *It is not recommended to use all four analysis times (0,6,12 and 18 UTC) per day and combine them with forecast fields to achieve 3-hourly temporal resolution. This causes unnecessary inconsistencies between 5h and 7h as well as 17h and 19h. The recommendation is to use 0 and 12 UTC and fill the times in between with forecast fields. I also thought that ERA-Interim on model levels were only available at 0,6,12 and 18UTC for public users, which gives me the indication that the access method was as a member-state user? Is this correct? Then you should have had access to ERA5 data all the time anyway.*

Thank you for correcting the information for ERA-Interim. We used the 0 and 12 UTC analysis times and filled the times in between with forecast fields as meteorological input data for FLEXPART.

"Temporally,  ERA-Interim has a resolution of 3 h, with  12 h analysis and 3 h forecast time steps." (L196−197)

Despite not being a member-state user, we modified the 'flex_extract' code to access the ERA-Interim data, which has a 3-hourly resolution with an analysis and forecast field mix in the full-access mode.

4. *After going through the manuscript I had a hard time to distinguish which data set and scheme/formula was used to calculate TE or scavenging coefficients. I would highly suggest to go through section 3.4 and 3.5 (below and in-cloud scavenging) again and try to be more clear in the description and distinguishing of where a scavenging coefficient comes from. Maybe by giving it different subscriptions.*

Please refer to the main response. We tried to improve the readability by unifying (replacing 'estimated' with 'measured') or differentiating ('simulated' and 'calculated') the terminology throughout the manuscript.

*Specific comments:*

1. *p.1 l.29: You mention diagnosing the scavenging coefficients from Flexpart. I thought that Flexpart defines the coefficients upfront in a species file. Therefore, I don't understand why the coefficients need to be diagnosed. Could you explain please?*

As you may already know, the species file (#40 for BC) contains the parameters for various efficiencies, such as the below-cloud collection efficiency for rain (pcrain_aero) and snow (pcsnow_aero) and in-cloud nuclei efficiency for cloud condensation (pccn_aero) and ice (pin_aero). The scavenging coefficients for in- and below-cloud are not included in the species file but are embedded in FLEXPART. Therefore, the purpose of our study, i.e., 'diagnosing the scavenging coefficients of FLEXPART', is necessary for evaluating BC accurately.

2. *p.2 l. 58: . . . because TE has been proven . . . . ; could you provide evidence*

We added a reason for the statement as follows:

"Accompanied with the refinement of BC emission inventories over East Asia (Choi et al., 2020; Kanaya et al., 2016), wet removal rates have been  a focal point to better predict BC behavior by using the term transport efficiency (TE), which is the observationally-determined fraction of undeposited BC particles during transport (e.g., Oshima et al., 2012; Kondo et al 2016), because TE shows a good relationship with accumulated precipitation along trajectory (APT; sum of precipitation over the past 72 h backward trajectory) (Choi et al., 2020; Kanaya et al., 2016) ." (L69−74)

3. *p.2 l. 65: what is meant by "mixing stats"?*

We apologize for the mistake; we intended to write 'mixing states,' which has been corrected.

"… the BC size distribution and mixing states during the spring of 2015 at the same location." (L81)

4. *p2. l.68-71: This sentence is hard to understand, especially the part with "...polluted areas scavenging in cloud were more...".*

The statement means that the pollution from East Asia might be more exposed to below- and/or in-cloud scavenging because high aerosol loading conditions are usually associated with significant cloud cover in East Asia. We added the following:

"… because high aerosol loading conditions are usually associated with  considerable cloud cover, which results in a higher frequency of wet scavenging (Eck et al., 2018)." (L85−87)

5. *p.2 l.73: . . . could be a useful parameter . . . ; I thought it is a useful parameter, why could?*

We revised this as follows:

"BC and carbon monoxide (CO) are byproducts of the incomplete combustion of carbon-based fuels, and the ratio between ΔBC (the difference from the baseline level) and ΔCO  is a useful parameter for characterizing  fuel types because of their different carbon contents (Zhou et al., 2009; Guo et al., 2017)." (L88−90)

6. *p.2 l.74: You mention that you adopt APT. You adopt it from where and how?*

We added a description of APT as follows:

"Accompanied with the refinement of BC emission inventories over East Asia (Choi et al., 2020; Kanaya et al., 2016), wet removal rates have been  a focal point to better predict BC behavior by using the term transport efficiency (TE), which is the observationally-determined fraction of undeposited BC particles during transport (e.g., Oshima et al., 2012; Kondo et al 2016), because TE shows a good relationship with accumulated precipitation along trajectory (APT; sum of precipitation over the past 72 h backward trajectory) (Choi et al., 2020; Kanaya et al., 2016) ." (L69−74)

7. *p.3 l.82: What are the administrative districts? Could you provide a plot?*

We revised the sentence and caption of Figure 1 as follows:

"The differences in wet removal rates depending on the measurement sites  and six administrative districts (Figure 1c), and season are discussed in Sect. 3.1 and 3.2, respectively." (L100−102)

"(c) The location of administrative districts and  spatial distribution of the mean BC mass in the potential emission region, which is the highest BC mass grid of each trajectory. The BC mass was obtained by multiplying (a) the emission rates and (b) the residence time."

8. *p3. l.85: Again, you estimate the scavenging coefficients from FLEXPART? Why? I sense that I might miss or misunderstand something.*

We replaced 'estimated from' with 'validated with' to clarify our research purpose. The wet scavenging coefficient is embedded in FLEXPART according to the pre-studied parameters. And the only way to verify those coefficients is a comparison between measured and simulated concentrations that comprises the uncertainty of emission inventories. However, for the first time, this study provides a more accurate assessment of scavenging coefficients that are widely used in many chemical models by excluding the uncertainty from emission inventories.

"…, the wet scavenging coefficients for below- and in-cloud processing were  validated with the measured wet removal rate by allocating the air mass location (such as below or within  cloud) and meteorological variables along the pathway of airmass transport." (L103−105)

9. *p.3 l.93: What does "intensive" in this context mean? Do you really mean intensive?*

The official name for those sites operated by NIER is 'Intensive Measurement Station'. We have changed the capital letters as follows:

"…, one of the Intensive Measurement Stations  operated by …" (L112−113)

10. *p.3 l.98: "The measurement periods were mainly in the early 2010s . . . "; Do you mean that they start in the early 2010s?*

As described in the caption of Figure S1, the measurement periods at three sites began in the early 2010s, e.g., 2010 for Baengnyeong, 2012 for Gosan, and 2011 for Noto. Despite the measurement periods being clarified in Figure S1, we added the measurement periods as follows:

"The longest measurement period was in Noto for approximately 6 years (from 2011 to 2016), followed by that in Baengnyeong (5 years; 2010 to 2017 except for 2011 to 2012) and Gosan (3 years; 2012 to 2015)." (L118-120)

11. *p.3 l.99: Since Figure S1 is in the supplement, you might want to add a note on that.*

Please refer to response #10.

12. *p.3 l.101: "well-validated" ; What is well-validated? There should be a criterion for this.*

We revised the sentence by adding the references for validation of the BC measurement as follows:

"In this study, we tried to obtain reliable BC concentrations from well-validated instruments, including OC–EC analyzers (Sunset Laboratory Inc., USA) with optical corrections, multi-angle absorption photometers (MAAPs; MAAP 5012, Thermo Scientific), and a continuous light absorption photometer (CLAP), yielding good agreement in the BC concentrations between the instruments (uncertainty $\leq \pm 15\%$, except for CLAP at $\leq \pm 20\%$) (Choi et al., 2020; Kanaya et al., 2008, 2013; Miyakawa et al., 2016, 2017; Taketani et al., 2016)." (L121−125)

"The overall uncertainty of  CO measurements from different instruments was estimated to be less than  5%,  which leads to a 10% uncertainty  in the overall regional ΔBC/ΔCO ratio (Choi et al., 2020)." (L136−138)

13. *p.4 l.118: Why mentioning GDAS?*

This statement clarified the use of ERA5 instead of GDAS, which is mainly used for HYSLPIT models, and emphasized that a much finer spatial resolution can provide more accurate backward trajectories compared to GDAS.

14. *p.4 l.119: Do the pressure levels correspond to ERA5 or GDAS?*

Yes, the pressure levels corresponded to GDAS, and those of ERA5 were described in L150. However, the pressure level of GDAS was deleted to improve the conciseness of the sentence as follows:

"..., as input for HYSPLIT instead of Global Data Assimilation System (GDAS; 1°×1° ) to improve the accuracy assessment of the airmass transportation pathway and to acquire more detailed information on the meteorological conditions." (L145−147)

15. *p.4 l.121: Did you disaggregate the precipitation fields as they are done for Flexpart simulations? For better comparison to Flexpart results.*

We used both precipitation fields, large-scale (lsp) and convective precipitation (cp) as described in L326, and we also added a description of precipitation as follows:

"According to the pathway of airmass transportation, the detailed meteorological information, such as  precipitation (sum of large-scale and convective precipitation),  clouds, and so on, was acquired from ERA5 hourly data at both single and pressure levels (37 levels; 1000 hPa to 1 hPa) according to the HYSPLIT backward trajectories ." (L148−153)

However, we did not interpolate the meteorological input data linearly to the position of computational particles in time and space (Hittmeir et al., 2018). Hittmeir et al. (2018) evaluated their new algorithms by comparing them with ECMWF 1 h data (0.5° resolution), which had a coarser resolution than used in the current study (0.25° resolution). In the case of HYSPLIT model, we considered the homogenous precipitation rate within a single grid cell because the temporal (1 h) and spatial resolution of ERA5 are dense enough to represent local variability compared to ERA-Interim (3 h with 1°). Moreover, the precipitation rates (lsp+cp) of HYSPLIT using ERA5 and FLEXPART using ERA-Interim showed negligible differences, thus justifying our results. A discussion of separating precipitation and uncertainty in the APT between the two datasets was included in Supplement S2 as follows:

"Figure S2 presents the cumulative probability of APT from HYSPLIT and FLEXPART. Although the airmass pathway showed insignificant differences between the two models, the median APT of FLEXPART (1.2 mm) was two times higher than that of HYSPLIT (0.63 mm), indicating a higher bias of the FLEXPART APT. This result can be caused by the difference in meteorological input data and the treatment of precipitation fields, homogeneous precipitation in a single grid cell (0.25°×0.25°) in HYSPLIT and disaggregated precipitation induced by interpolating in time and space in FLEXPART (Hittmeir et al., 2018). The higher bias in the FLEXPART APT contributed to increasing the magnitude of the underestimation of FLEXPART TE when assuming the same APT from HYSPLIT model, indicating an insignificant impact on the results." (Sect. S2 in the Supplement)

[Figure]

**Figure S2**. Cumulative probability plot of the APT from HYSPLIT (blue) and FLEXPART (red) during the study periods at the three sites. The dashed black line indicated a cumulative probability at 0.5 (median).

16. *p.4 l.124: What are the main BC regions? How is "main BC region" defined?*

The 'main BC source region' was intended to represent the highest BC-emitting region during airmass transport. We added a detail description of the main BC source region as follows:

"As the airmass was being transported, if precipitation occurred before the airmass arrived at the main BC source region, which is the highest BC emission area,  then the magnitude of wet removal effect as a function of APT could be underestimated at receptor sites because the airmass containing BC would not have been exposed to wet scavenging conditions ." (L154−157)

17. *p.4 l.125: Why couldn't the precipitation not be overestimated?*

This statement indicated the underestimation of wet scavenging by precipitation and not the amount of precipitation. If precipitation occurred before arriving at the main BC source region, the airmass containing BC would not have experienced wet scavenging, which may be misinterpreted as a reduced impact of wet scavenging. Moreover, Nogueira (2020) reported that ERA5 showed lower bias and higher correlations with Global Precipitation Climatology Project (GPCP) compared to ERA-Interim at the mid-latitude regions. Please refer to our response to comment #16.

Nogueira, M.: Inter-comparison of ERA-5, ERA-interim and GPCP rainfall over the last 40 years: Process-based analysis of systematic and random differences, Journal of Hydrology, 583, 124632, https://doi.org/10.1016/j.jhydrol.2020.124632, 2020.

18. *p.5 l.150: What do the global models have to do with this study?*

The relevant sentence explained the reason for excluding the dry deposition effects in our analysis because the our estimated dry deposition velocity was even smaller by a factor of 3−10 compared with that used in various global models (such as NCAR CAM3, GISS GCM II-prime, MOZART-4, and so on). We revised the relevant sentence to specify the dry deposition velocity in global models and provided references as follows:

"Although TE is also affected by dry deposition,  Choi et al. (2020) reported that the effect of dry deposition could be  neglected because dry deposition velocities (0.01−0.03 cm s$^{-1}$) are much lower than the default setting (0.1 cm s$^{-1}$) in global models (Chung and Seinfeld, 2002; Cooke and Wilson, 1996; Emmons et al., 2010; Sharma et al., 2013)." (L184-187)

19. *p5. l.155: shouldn't there be a reference to the Flexpart v10.4 paper?*

We added Pisso et al. (2019) with Stohl et al. (2005). (L191)

20. *p.5 l.159: ERA-Interim is not an operational reanalysis. ERA-Interim was suspended and ERA5 is now operational!*

Thank you for correction. We deleted term of 'operational' as follows:

"The FLEXPART model was executed with  reanalysis meteorological data from the ECMWF ERA-Interim at a spatial resolution of 1°×1° with 60  model levels from surface up to 0.1 hPa." (L195−196)

21. *p.5 l.159-160: You should rewrite to "60 model levels" since vertical levels could also be pressure levels or others.*

We revised the expression according to the reviewer's suggestion.

"The FLEXPART model was executed with  reanalysis meteorological data from the ECMWF ERA-Interim at a spatial resolution of 1°×1° with 60  model levels from surface up to 0.1 hPa." (L195-196)

22. *p. 5 l.160: "ECMWF has a resolution of 3h. . ." this is wrong. The ERA-Interim data set has this resolution, but not ECMWF in general.*

We revised the sentence as follows:

"Temporally,  ERA-Interim has a resolution of 3 h, with  12 h analysis and 3 h forecast time steps." (L197−198)

23. *p.5 l.168: what do you mean by "extracted" ? How do you calculate the TE with Flexpart data.*

By modifying the FORTRAN code (wetdepo.f90) to print out the wet scavenging coefficients of each grid cell, TE can be calculated from equations (2) and (3) in Sect. 3.3. To enhance the readability, we referred to Sect. 3.3, which contains more detailed information as follows:

"To validate the wet scavenging scheme in FLEXPART by comparison with the measured TE value, the wet scavenging coefficients for below- and in-clouds were extracted from FLEXPART to calculate TE (see Sect. 3.3 for more details)." (L204−206)

24. *p.5 l.174: Could you define the bins somewhere? This should be done according to be able to reproduce the results.*

We added a description of APT bins in the caption of Figure 2 as follows:

"The 9 bins consist of 0.01–0.25, 0.25–0.50, 0.50–0.75, 0.75–1.0, 1.0–2.5, 2.5–5.0, 5.0–10, 10–20, and 20–30 mm."

25. *p.5 l.178: What was R2 before it was improved?*

We added the $R^2$ value (0.940) from the widely used equation $A–B\times log(APT)$ as follows:

"…, because the coefficients of determination ($R^2$) was improved underline{from 0.940}  to 0.981 though TE values from three sites were used (Table 1)." (L219−220)

26. *p.5 l.183: This Fukue site comes out of nowhere and it is not clear where it is located.*

The results from Fukue were from a previous study; thus, we added the longitude and latitude of Fukue site as follows:

"The parameters $A_1$ (0.269 ± 0.039) and $A_2$ (0.385 ± 0.035) of the overall fitting were higher and lower, respectively, than the derived equation from the Fukue site ($A_1 = 0.109$ and $A_2 = 0.68$), which is a remote site in Japan (128.68º E, 32.75º N) (Kanaya et al., 2016)." (L222-224)

27. *p.6 l.187: I don't understand how the new SED indicates the transport to the Artic, please explain further.*

Zhu et al. (2020) reported that the anthropogenic BC emitted from East Asia and Russia could contribute significantly to Arctic BC at surface level (62%; 56% for Russia and 6% for East Asia) and high altitudes (48%; 8% for Russia and 40% for East Asia) according to the FLEXPART model. The statement 'transport to the Arctic' means that the contribution of BC emitted from East Asia (also Russia) can be affected by the wet scavenging efficiency and our new derived SED indicated that reduced scavenging efficiency resulted in more transport of BC to the Arctic compared to Kanaya et al. (2016) "because $A_2$ determines the magnitude of the wet removal efficiency according to APT" (L229). We revised the relevant sentence to more clearly convey our intended meaning as follows:

"In particular, the $A_2$ value is important for calculating the  amount of BC from emission sources via  long-range transport, e.g., toward the Arctic (Kanaya et al., 2016; Zhu et al., 2020), because $A_2$ determines the magnitude of the wet removal efficiency according to APT. Thus, the newly obtained SED equation, which has a low $A_2$ value, indicates that more BC  might be transported to the Arctic region than  that reported by Kanaya et al. (2016)." (L226−230)

28. *p.6 l.214: what global model?*

We revised the sentence as follows:

"This calculated *e*-folding lifetime in East Asia was much shorter than 16.0 days for BC from FLEXPART v10 (Grythe et al., 2017)." (L255−257)

29. *p.7 l.224: " A similar tendency of R2 , TE=0.5 also showed .. " ; I don't understand this formulation. Do you mean R2 and TE ?*

We revised the sentence as follows:

"A similar tendency of $R^2$, the APTs to achieve TE=0.5 also showed regional differences , i.e., higher in East and North China and lower in other regions." (L267−268)

30. *p.8 l.258: wasn't this described in Grythe et al. 2017?*

Grythe et al. (2017) also discussed $f_g$, although they did not include a detailed equation and constants for calculating $f_g$. Therefore, we only cited the study by Stohl et al. (2005).

31. *p.8 l.258 – 261: Its not only the grid resolution but the whole model physics is different apart from the differences between Hysplit and Flexpart. Additionally, regional/local pattern of precipitation*

*and clouds are totally different especially because Flexpart uses disaggregated precipitation while it seems that Hysplit and ERA5 data used in this paper study weren't disaggregated. How does this reflect in the results? And why did you chose 1°x1° for ERA-Interim instead of the 0.75°x0.75° resolution ERA-Interim was stored on?*

Please refer to our responses to comments #2 and #15. We used the default setting of 'flex_extract' (FpExtractEcmwfData-7.0.2.tar.gz; available from https://www.flexpart.eu/downloads); 1°×1° for ERA-Interim was the only option at that time.

32. *p.8 l. 274.: In this section, I got confused by the values from measured vs ERA5 vs calculated vs reported vs Flexpart. Did you calculate the TE with the Flexpart scheme from ERA5 data? Then, what did you use from Flexpart simulation results?*

Please refer to the main response and comment #4 in major comments.

33. *p.8 l.301: again, could you please define the bins somewhere? (reproducabilty)*

We added a description of precipitation rate bins in the caption of Figure 5 as follows:

"The 11 bins consist of 0.01–0.04, 0.04–0.06, 0.06–0.08, 0.08–0.1, 0.1–0.2, 0.2–0.4, 0.4–0.6, 0.6–0.8, 0.8–1, 1–2, and 2–3 mm $hr^{-1}$."

34. *p.8 l.307: what are reported values?*

We revised the sentence as follows:

"Figure 6 shows the comparison of $\Lambda_{below}$  calculated using equations from previous studies with  that derived using our equation by assuming that the BC size was approximately 200 nm." (L377−378)

35. *p.8 l.319: could you give a reference for your statement of " the effect of differences in diameter might be negligible" please?*

When we applied a larger diameter close to that of secondary ions (~ 500 nm) to the equation from Laakso et al. (2003), $\Lambda_{below}$ increased only 30% compared with that of ~200 nm (Figure R1), resulting in a similar average MFB (from 1.64 to 1.54) with Xu et al. (2017). Thus, we added the following:

"For example, the $\Lambda_{below}$ of secondary ions in Beijing (Xu et al., 2017) had the highest MFB (1.68), and although the diameter ranges were larger (~ 500 nm) than those of BC, the effect of differences in diameter might be negligible because significant difference in $\Lambda_{below}$ between two diameters were not observed (less than 30%) when applied to Laakso et al. (2003)." (L389−392)

[Figure]

**Figure R1.** Same as Figure 6 except for adding the results obtained by applying a larger diameter (500 nm) instead of 200 nm in the equation from Laakso et al. (2003).

36. *p.10 l.333: the Flexpart scavenging coefficient is taken from the simulations with ERA-Interim data and the estimated coefficient is from the measurement data in combination with the scheme from Flexpart and ERA5 data? Is this correct?*

That is not correct. The FLEXPART scavenging coefficients ($\Lambda$; now 'calculated' $\Lambda$) were calculated by HYSPLIT with ERA5 pressure level data according to the FLEXPART scheme. The estimated (now 'measured') $\Lambda_{in}$ was calculated from the measured TE with HYSPLIT and ERA5. As mentioned in our main response, only the measured TE was compared with the simulated TE

from FLEXPART, and the measured below- and in-cloud scavenging coefficients were compared with those calculated according to the FLEXPART scheme (not simulated).

*37. p10. l.362: what would be the effect if it would be 4:1 or 2:1?*

When the training and test sample ratio was varied from 4:1 to 2:1, the relative importance in Figure 7b changed slightly, but CAPE was still the most important variable among the six input variables.

*38. Table 2: Does this mean Flexpart does not correspond to the ERA-Interim simulation results but to the calculated values with ERA5 data and the Flexpart scheme? If not, reformulate please.*

Please refer to the main response and comment #4 in major comments.

We revised the caption of Table 2 to clarify the meaning of $\Lambda_{below}$ and $\Lambda_{in}$ and replaced 'FLEXPART' with 'calculated.'

**Table 2.** Summaries of the transport efficiency (TE) and scavenging coefficients for selected (a) below- and (b) in-cloud cases based on ERA5 hourly data of pressure levels from ECMWF.

| Cases | Median | Interquartile range (25th percentile – 75th percentile) |
|---|---|---|
| (a) Below cloud ($N_{case}$=831) | | |
| TE | 0.89 | [0.61 – 1.27] |
|  Measured $\Lambda_{below}$ (s$^{-1}$) | $4.01\times10^{-6}$ | [$2.70\times10^{-6} - 6.33\times10^{-6}$] |
|  Calculated $\Lambda_{below}$ (s$^{-1}$) [a] | $6.63\times10^{-6}$ | [$6.38\times10^{-6} - 7.08\times10^{-6}$] |
| | | |
| (b) In-cloud ($N_{case}$=769) | | |
| TE | 0.72 | [0.43 – 1.06] |
|  Measured $\Lambda_{in}$* (s$^{-1}$) [b] | $8.06\times10^{-5}$ | - |
|  Calculated $\Lambda_{in}$* (s$^{-1}$) [a, b] | $7.28\times10^{-6}$ | - |

[a] Calculated using the FLEXPART scheme
[b] Overall median value

*Technical corrections:*

*p.3 l.83: Afterward → Afterwards*

This has been corrected. (L102)

*p.5 l.150: remove "but"*

This has been corrected. (L185)

*p.5 l.150: negligible → neglected*

This has been corrected. (L185)

*p.5 l.153: I would suggest to exchange the order of "model simulation" and "measured values"*

This has been corrected. (L189)

*p.5 l.166: were existed → were available*

This has been corrected. (L204)

*p.7 l.223: was varied → varied*

This has been corrected. (L266)

*p.7 l.230: " than the dominat in-cloud . . ." → than in the dominant ...*

We revised it as follows:

"...; therefore, the wet removal efficiency should be lower than that in the dominant in-cloud scavenging region." (L273−274)

*p.7 l.249: simulation → simulations*

This has been corrected. (L317)

[revised manuscript text omitted]

**S1 Uncertainty in the transport efficiency (TE) and below- and in-cloud scavenging coefficients**

Our main results, including the TE, $\Lambda_{below}$, and $\Lambda_{in}$, could be influenced by selecting (1) different starting altitudes of the backward trajectories and (2) different altitude criteria for identifying the potential emission region.

First, to investigate the uncertainty caused by different starting altitudes of the backward trajectories, we analyzed the Welch's $t$-test for APT derived from starting altitudes of 500 m and 1000 m. The APT between the two datasets did not show a significant difference (3%) ($p \geq 0.1$). Depending on the site, the TE showed a significant difference ($p < 0.05$) at Gosan only at a relatively small value of −4.2%. In the case of regional TE, Northeast China and South Korea were significantly different ($p < 0.01$), with original values up to −15%; however, the corresponding APT for achieving TE=0.5 and TE=1/$e$ only decreased by −6% and −2%, respectively. The regional wet removal efficiency was more apparent, such as more or less APT needed to attain TE=0.5 and TE=1/e in low-efficiency regions (East and North China) and high-efficiency regions (South Korea and Japan), respectively. For the high starting altitude, i.e., 1000 m, the airmass had a higher chance of being exposed to in-cloud scavenging resulting in a much lower TE for in-cloud scavenging (−3%). Otherwise, the TE for below-cloud scavenging cases was increased by 7% because of a reduced chance to expose washout effects (Table S1). Because of the variations in the TE for below- and in-cloud scavenging cases, the calculated median $\Lambda_{below}$ and $\Lambda_{in}$ converged within a similar range as the original results. It should be noted that the median measured $\Lambda_{below}$ was slightly higher than the calculated $\Lambda_{below}$ according to FLEXPART, which is opposite the original results. The small difference could be ignored when considering the insufficient sample number for below-cloud cases at a starting altitude of 1000 m.

Second, we also checked the difference in wet scavenging efficiency, which can be caused by applying 1.5 km (instead of 2.5 km) as a threshold to determine the potential emission region. The identified six administrative districts for potential emission regions at an altitude of 1.5 km were same as those at an altitude of 2.5 km. The median traveling time from potential source regions to receptor sites was decreased from 38 h to 25 h when precipitation occurred because the individual potential source region was closer to the receptor site because the selection altitude was decreased. However, the difference in traveling time did not significantly influence our final results because the TE for below- and in-cloud cases only decreased by 1% and 6% and the measured $\Lambda_{below}$ and $\Lambda_{in}$ were consistent with the original results within ±54% (Table S2). From these results, we confirmed the representativeness of our regional and seasonal wet removal efficiency analysis.

**S2 Difference in airmass pathways and accumulated precipitation along trajectory (APT) between HYSPLIT and FLEXPART**

We investigated the uncertainty in the airmass pathway and APT between HYSPLIT model using ERA5 and FLEXPART using ERA-Interim during study periods at three sites. It should be noted that the trajectory of FLEXPART was selected as the center of the main grid (1°×1°) according to the highest residence time in the same time interval and then compared with HYSPLIT results by calculating the distance between two hourly endpoints. Thus, differences of less than ~100 km can be regarded as a good agreement when considering the grid resolution of FLEXPART. The difference in distance increased as the traveling time was increased. However, the median traveling time of airmasses, including APT=0 case, was 31 h, which showed a difference in distance of ~100 km. When the traveling time was expanded up to the 75%ile of the traveling time (50 h), the difference in distance was close to ~200 km. Although the difference in distance at 72 h traveling time was high, 72 h traveling time cases was so rare that we could neglect the impacts on our results. In total, the median difference in distance was ~47 km, suggesting good agreement between the two datasets. In addition, the difference in accumulated backward-trajectory endpoints was much smaller because random errors in the single calculations can be diminished by increasing the number of calculations (Gebhart et al., 2005; Jeong et al., 2017).

Figure S2 presents the cumulative probability of APT from HYSPLIT and FLEXPART. Although the airmass pathway showed insignificant differences between the two models, the median APT of FLEXPART (1.2 mm) was two times higher than that of HYSPLIT (0.63 mm), indicating a higher bias of the FLEXPART APT. This result can be caused by the difference in meteorological input data and the treatment of precipitation fields, homogeneous precipitation in a single grid cell ($0.25° \times 0.25°$) in HYSPLIT and disaggregated precipitation induced by interpolating in time and space in FLEXPART (Hittmeir et al., 2018). The higher bias in the FLEXPART APT contributed to increasing the magnitude of the underestimation of FLEXPART TE when assuming the same APT from HYSPLIT model, indicating an insignificant impact on the results.

Gebhart, K.A., Schichtel, B.A., Barna, M.G.: Directional biases in back trajectories caused by model and input data. J. Air Waste Manag. 55, 1649-1662, 2005.

Jeong, U., Kim, J., Lee, H., Lee, Y.G.: Assessing the effect of long-range pollutant transportation on air quality in Seoul using the conditional potential source contribution function method. Atmos. Environ., 150, 33-44, 2017.

Hittmeir, S., Philipp, A., and Seibert, P.: A conservative reconstruction scheme for the interpolation of extensive quantities in the Lagrangian particle dispersion model FLEXPART. Geosci. Model Dev., 11, 2503-2523, https://doi.org/10.5194/gmd-11-2503-2018, 2018.

[Figure]

**Figure S1.** Time series of the ΔBC/ΔCO ratio and accumulated precipitation along trajectory (APT) during the measurement periods in (a) Baengnyeong (1 Jan 2010–31 Dec 2016), (b) Gosan (1 May 2012–30 Apr 2015), and (c) Noto (1 Jan 2011–31 Dec 2016). The square symbols with solid lines indicate monthly concentrations. The red shaded region in the Baengnyeong figure indicates periods of data missing from 2011 to 2012 due to the absence of CO data.

[Figure]

**Figure S2.** Cumulative probability plot of the APT from HYSPLIT (blue) and FLEXPART (red) during the study periods at the three sites. The dashed black line indicated a cumulative probability at 0.5 (median).

**Table S1.** Same as Table 2 except for the different backward trajectory starting altitudes (1000 m)

| Cases | Median | Interquartile range (25th percentile – 75th percentile) |
|---|---|---|
| (a) Below cloud ($N_{case}$ = 262) | | |
| TE | 0.95 | [0.65 – 1.28] |
| Measured $\Lambda_{below}$ (s$^{-1}$) | $8.85 \times 10^{-6}$ | [$6.57 \times 10^{-6}$ – $1.46 \times 10^{-5}$] |
| Calculated $\Lambda_{below}$ (s$^{-1}$) [a] | $7.49 \times 10^{-6}$ | [$6.83 \times 10^{-6}$ – $8.42 \times 10^{-6}$] |
| | | |
| (b) In-cloud ($N_{case}$ = 953) | | |
| TE | 0.70 | [0.46 – 1.02] |
| Measured $\Lambda_{in}$* (s$^{-1}$) [b] | $7.67 \times 10^{-5}$ | - |
| Calculated $\Lambda_{in}$* (s$^{-1}$) [a,b] | $8.01 \times 10^{-6}$ | - |

[a] Calculated using FLEXPART scheme
[b] Overall median value

**Table S2.** Same as Table 2 except for the different altitude criteria (1.5 km) for identifying potential emission source regions.

| Cases | Median | Interquartile range (25th percentile – 75th percentile) |
|---|---|---|
| (a) Below cloud ($N_{case}$ = 436) | | |
| TE | 0.88 | [0.60 – 1.24] |
| Measured $\Lambda_{below}$ (s$^{-1}$) | $6.17 \times 10^{-6}$ | [$2.55 \times 10^{-6}$ – $1.39 \times 10^{-5}$] |
| Calculated $\Lambda_{below}$ (s$^{-1}$) [a] | $7.52 \times 10^{-6}$ | [$6.88 \times 10^{-6}$ – $8.50 \times 10^{-6}$] |
| | | |
| (b) In-cloud ($N_{case}$ = 282) | | |
| TE | 0.68 | [0.44 – 1.03] |
| Measured $\Lambda_{in}$* (s$^{-1}$) [b] | $9.39 \times 10^{-5}$ | - |
| Calculated $\Lambda_{in}$* (s$^{-1}$) [a,b] | $8.15 \times 10^{-6}$ | - |

[a] Calculated using FLEXPART scheme
[b] Overall median value